# A prefrontal motor circuit initiates persistent movement

Yihan Wang[1,2] & Qian-Quan Sun [1,2,3] ✉

Persistence reinforces continuous action, which benefits animals in many aspects. Diverse external or internal signals may trigger animals to start a persistent movement. However, it is unclear how the brain decides to persist with current actions by selecting specific information. Using single-unit extracellular recordings and opto-tagging in awake mice, we demonstrated that a group of dorsal mPFC (dmPFC) motor cortex projecting (MP) neurons initiate a persistent movement by selectively encoding contextual information rather than natural valence. Inactivation of dmPFC MP neurons impairs the initiation and reduces neuronal activity in the insular and motor cortex. After the persistent movement is initiated, the dmPFC MP neurons are not required to maintain it. Finally, a computational model suggests that a successive sensory stimulus acts as an input signal for the dmPFC MP neurons to initiate a persistent movement. These results reveal a neural initiation mechanism on the persistent movement.

Medial prefrontal cortex (mPFC) regulates decision-making by amplifying certain information while suppressing others[1–3]. In the mPFC, motor projecting (MP) neurons majorly projects to the primary motor neurons in almost all motor cortices and the striatum, but less to other deep brain regions and local non-MP neurons[4]. Therefore, MP neurons may be involved in the most downstream mPFC circuit, which collects all filtered information to affect subsequent behavior. As such, we asked (Q1) if the MP neurons play a role on instructing subsequent movement in decision-making.

Contextual information includes internal beliefs about external salient and physiological states[5]. Valence in this study refers to the natural[6], but not learned[7], evaluation of external stimulation. In the multi-liquid licking task, the sweeter they tasted, the more frequently the mice licked[8]. It is thus possible that the licking movement is controlled by valence information. However, when the mice licked persistently, they showed a similar licking frequency in the water and sucrose sessions (Supplementary Fig. 1e, f). The question therefore arises as to how the brain uses contextual or valence information to initiate a persistent movement. Since the MP neurons receive unidirectional inputs[4] from the insular cortex (IC), which encodes valence[8], and from the basal lateral amygdala (BLA), which is responsible for valence assignment[7] (internal belief), we asked (Q2) what type of information (contextual or valence) the MP neurons encode during a persistent movement.

In the mPFC, the MP neurons are divided into two subtypes: dmPFC MP and vmPFC MP[4]. They may have different functions. Different from vmPFC[9], the neurons in the dmPFC is associated with memory retrieval[10] and executive function[11]. Thus, it is more likely for dmPFC MP neurons to lead a persistent movement. Therefore, we only examined the MP neurons in the dmPFC here.

This study answered the two questions mentioned above. For Q1, we found that dmPFC MP neurons are responsible for the initiation of a persistent movement, but they are not involved in the information processing before the movement onset and do not control movement kinematics at the subsequent maintenance. For Q2, we found that the dmPFC MP neurons encode contextual information rather than valence during the initial phase of a persistent movement. Our results suggest that animals tend to ignore valence during the initial phase of a persistent movement and that initiation is driven by MP neurons in the dmPFC.

[1]Graduate Neuroscience Program, University of Wyoming, Laramie, WY 82071, USA. [2]Department of Zoology and Physiology, University of Wyoming, Laramie, WY 82071, USA. [3]Wyoming Sensory Biology Center of Biomedical Research Excellence, University of Wyoming, Laramie, WY 82071, USA. ✉e-mail: neuron@uwyo.edu

## Results

### Behavioral quantification

First, we defined a persistent movement as the continuous repetition (>6 Hz) of a single movement (e.g., a cycle of tongue or limb movements) and the maintenance of that continuity for at least 5 seconds (see non-persistent movement, Supplementary Movie 1, vs persistent movement, Supplementary Movie 2). To impose a persistent movement on mice, they were deprived of water (Health data is given in the Behavioral details, Methods) and then head-fixed to a custom-made set-up and trained to lick various types of liquid after delivery onset (DO) (Fig. 1a, Supplementary Fig. 1b, and Behavioral details, Methods) in the darkness, but received no other artificial stimuli. Licking signals, facial and locomotor activities were measured. After training, we observed that the mice showed licking movements sustained for approximately 15-30 seconds during water delivery (Supplementary Fig. 1c–f). Consistent with the standard pattern of affective dynamics performed previously[12], licking frequency was maximized in the initial phase (Supplementary Fig. 1c–f, right column, peaks of licking frequency are indicated by black arrows) and then stabilized at 6 to 7 Hz until the end of delivery (Supplementary Fig. 1d, e) or when the liquid was switched to quinine (5 mM, Supplementary Fig. 1c, f). Higher hedonic stimuli, 20% sucrose, did not increase this frequency (Supplementary Fig. 1e, f). As an aversive stimulus, quinine administration was more likely than the interruption of water or sucrose administration to cause the termination of persistent lick ($p < 0.05$ with respect to termination bias, Supplementary Fig. 1i). In addition, we found no significantly behavioral difference between male and female mice in the initiation and termination of persistent licking movement (Supplementary Fig. 1r, s). Therefore, quinine was used to terminate the persistent licking movement and to evaluate negative valence; by

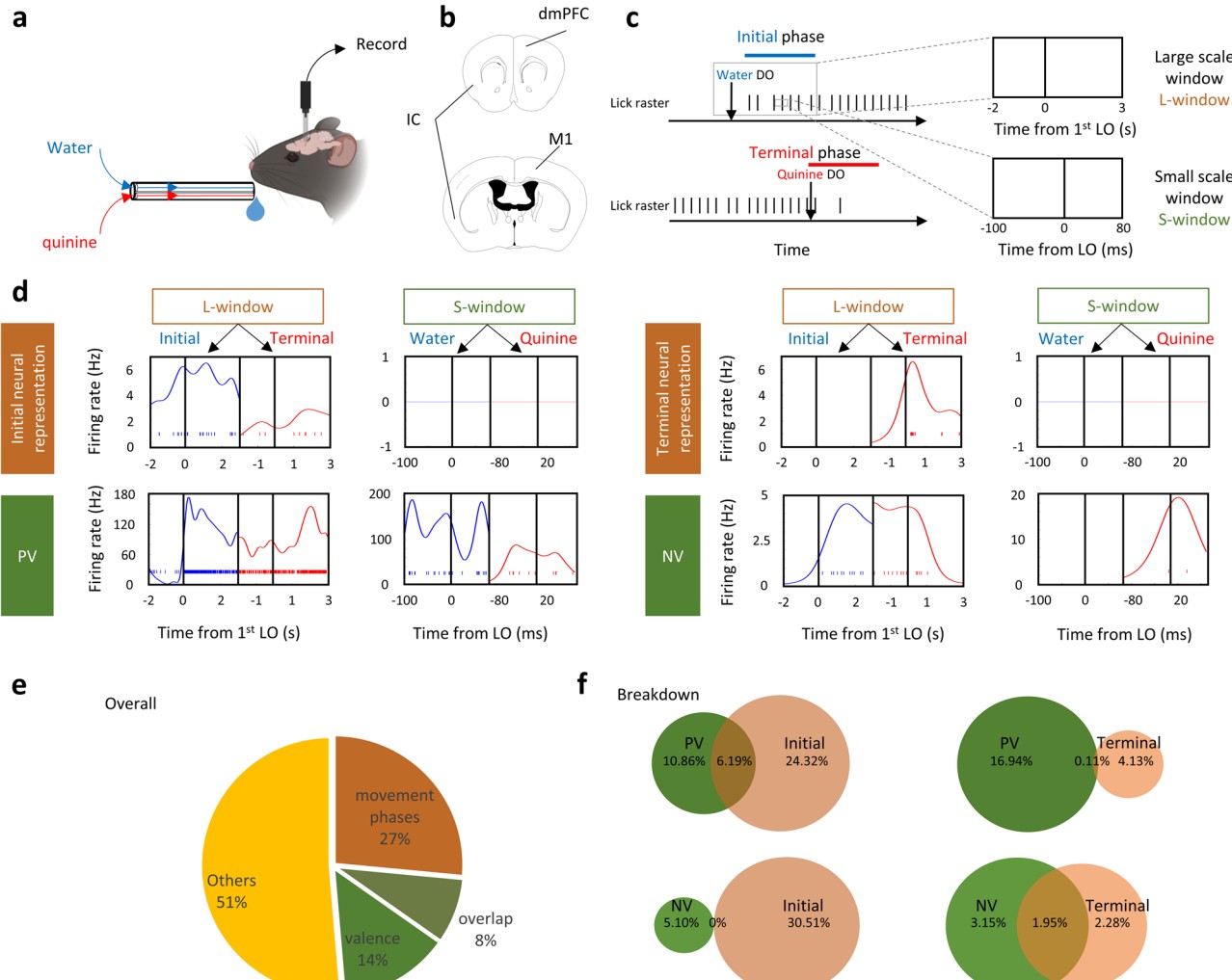

**Fig. 1 | Dissociable neural representations of movement phases and valence. a** Schematic of the experimental setup. Water or quinine was continuously delivered through a multi-lick-ports with a time delay of zero at the delivery switches. Mice were trained to respond the liquid delivery without conditioning. Created with BioRender.com under a Creative Commons Attribution-NonCommercial-NoDerivs 4.0 International license https://creativecommons.org/licenses/by-nc-nd/4.0/deed.en. **b** Schematic of recording sites. Used with permission of Elsevier Science & Technology Journals, from The Mouse Brain in Stereotaxic Coordinates, Keith et al, edition 3, 2008; permission conveyed through Copyright Clearance Center, Inc. IC, insular cortex; dmPFC, dorsal medial prefrontal cortex; M1, primary motor cortex. **c** A representative task of persistent lick. Two recording windows were selected near the water or quinine delivery onset (DO). For the large scale window (L-window), the recording epoch began 2 seconds before the 1st water or quinine lick onset (LO) until 3 seconds after the first water or quinine LO. For the small scale window (S- window), the recording epoch began from 100 ms before the LO to 80 ms after the LO. **d** Spike raster and firing rate estimation of four representative single- units classified to represent the initial phase and terminal phase of persistent movement, positive valence (PV), and negative valence (NV) in the L- window and S- window, respectively. Spike raster and firing rate under water and quinine licks are colored blue and red, respectively. **e** Pie chart showing the percentage of the classified neural representations. **f** Venn diagram of the neural representations of valence and movement phases. Percentages of each category in all recorded neurons are shown in the diagram.

contrast, water or sucrose delivery was used to trigger licking movement and the taste of them was used to evaluate positive valence.

Next, we determined the temporal window for the study of valence and movement phases. For valence, the assessment of whether water or quinine is hedonic or aversive should be based on the single contact (licking onset (LO) ± 180 ms, including 100 ms contact time with liquid before the tongue touches the lick port and the subsequent 80 ms, Supplementary Fig. 1c–f) to them. For movement phase, the maximum licking frequency was confined within the initial phase and a sharp decrease of licking frequency was in the terminal phase (Supplementary Fig. 1c–f), so we suspected that there should be additional neural signal (contains contextual information) to increase and decrease licking frequency besides licking command. Based on the period of peak licking frequency (approximately 3 s, as indicated by the arrow, Supplementary Fig. 1c) in the initial phase and the sharp drop of licking frequency (approximately 3 s from 6-7 Hz to less than 1 Hz, as indicated by the arrow, Supplementary Fig. 1c) in the terminal phase, we designed a time window of 5-second long (including 2 s baseline before the licking onset) as a temporal window for data analysis (Fig. 1c).

## Dissociable neural representations of movement phases and valence

To characterize the single-units that statistically represent valence and movement phases, we used in vivo silicon probe to collect neuronal activity data from all three brain regions, including the insular cortex (IC), which is known to encode valence in the gustatory system[6,8], the primary motor cortex (M1), which represents the motor commands of voluntary movement[13], and the dorsal medial prefrontal cortex (dmPFC), which has been shown to connect these two brain regions[4]. The single-units that were generated from the neural activity data were then classified into different groups of neural representations according to the extent to which they can discriminate liquid types (for valence) or movement phases (Methods).

We then asked whether the neural representation of movement phases and valence are associable or not by examining the proportion of neural representations of valence and movement phases in three brain regions (IC, M1, and dmPFC). As population, we found that 27% of single-units represented movement phases (initial or terminal phase, ≥65% in movement phase correlation) but showed weak or no taste tuning (Fig. 1d-f). 14% of single-units represented valence (positive or negative valence (PV or NV), $z > 1.64$ in taste correlation) but exhibited poor specification on movement phases (Fig. 1d-f). By contrast, only small fraction (8%) of single units displayed the preference to both valence and movement phases (Fig. 1e, f), though this number varied trivially from region to region (Supplementary Fig. 4).

To test whether the neural networks representing movement phases and valence interact with each other, we examined the connectivity between the neurons solely tuned to initial phase and PV using Total Spiking Probability Edges (TSPE)[14] and compared it with shuffled connectivity. Our results showed that there is no overall excitatory impact from the initial phase to PV or from PV to initial phase tuned neurons (mean of real TSPE < 99 percentile of shuffled TSPE, Supplementary Fig. 5d). Note that connectivity between the terminal phase and NV tuned neurons was not available because their spike data were unable to construct a cross-correlation in a 50ms-time window, suggesting that the coding of terminal phase and NV are not connected. Together, these results suggest that the movement phases, which contain generally contextual information, and valence are encoded separately during a persistent movement.

## The representation of initial phase in dmPFC MP neurons

Next, we compared the fraction of clustered neural representations in the IC, M1, and dmPFC (Supplementary Fig. 4). The results confirmed that licking command were enriched in the M1 (licking representations

in M1 vs dmPFC or IC = 12% vs 5% or 5%; Supplementary Fig. 4a1, b1, c1) and dissociable coding of valence and movement phases (Supplementary Fig. 4a3, b3, c3). Then, we examined decoding performance in these brain regions by training a linear discriminant decoder on firing rate data (Supplementary Fig. 6 and 7). We noticed the stable encoding of valence in the IC whereas poor specification in the M1 (Supplementary Fig. 6a, b). Interestingly, dmPFC neurons showed gradually faded coding of valence across lick trials (Supplementary Fig. 6c). Moreover, the highest decoding accuracy of movement phases was showed in the dmPFC among IC, M1, and dmPFC. Given the unidirectional information flow IC -> mPFC -> M1[4], we speculated that the valence and contextual signal could be modified in the dmPFC. Since MP neurons in the dmPFC directly connect M1, we examined what information they contained. Using cell-specific recordings enabled by the opto-tagging approach[15] (Fig. 2a-c), we found that MP neurons in the dmPFC showed good representation of the initial phase, while selectivity for PV and NV was low. Specifically, the neuronal clustering results showed that 32% of dmPFC MP neurons exhibited a degree (≥65% shuffled activity at initial phase & ≤35% shuffled activity at terminal phase) of initial phase representation (Fig. 2h), whereas only 3% of them exhibited a degree (≥65% shuffled activity at terminal phase & ≤35% shuffled activity at initial phase) of terminal phase representation (Fig. 2h). No more than 20% of dmPFC MP neurons showed a degree ($z > 1.64$ in valence correlations compared to shuffled activity) of valence representation (PV + NV, Fig. 2e). The decoding performance results showed that dmPFC MP neurons had a low ability to discriminate positive and negative valence ($p < 0.0001$ lower than IC, Fig. 2k left), while had a high representation of movement phases ($p < 0.0001$ higher than IC and shuffled cumulative decoding accuracy, Fig. 2k right). To confirm the discriminability of dmPFC MP neuron on valence and movement phases, we first embedded neuronal population activity in the S- and L-window of dmPFC MP neurons into trajectories using principal component analysis (PCA) and then measured the mean Euclidean distances in all PC dimensions. The results showed that the discrimination of neural activity was significant higher in the L-window than it in the S-window ($p < 0.05$, Fig. 2j), indicating that dmPFC MP neurons discriminate better between movement phases than between valence values. This was different from whole dmPFC neurons, which encode both movement phases and valence (Supplementary Fig. 6c and Supplementary Fig. 7c) in the initial phase of a persistent movement. Moreover, this movement phases coding emerged after the LO (Fig. 2l), suggesting that dmPFC MP neurons are not involved in the process of information filtering. Overall, our results suggest that the valence signal is filtered out in the dmPFC when it is transmitted from the dmPFC MP neurons to the motor cortex during the initial phase of persistent movement.

## Effects of silencing dmPFC MP neuron on persistent movement

To test the role of dmPFC MP neuron on persistent licking movement in different phases, we examined the initiation and termination bias, as well as licking frequency (Methods). dmPFC MP neurons were optogenetically manipulated by expressing stGtACR2[16] and shining laser during the different phases of the persistent licking task (Fig. 3a). Our results showed that optogenetic silencing of dmPFC MP neurons impaired the initiation of licking ($p < 0.001$ compared to the sham trials, Fig. 3b, c), but had no effect on the termination of lick ($p > 0.05$ compared to the sham trials, Fig. 3d, e) or the licking frequency in the middle phase (Fig. 3h, i) in thirsty mice. It suggests that dmPFC MP neurons are functional as movement initiation but not involved in the control of individual licks, which conducted by the primary motor neurons[17,18]. The similarity of thirst level (by comparing body weight loss), facial activities, and locomotor activities were confirmed in sham and laser trials (Supplementary Fig. 9). To collect parallel evidence for the function of dmPFC MP neuron in initiating persistent licking movement, dmPFC MP neurons, expressing hm4D(Gi), were

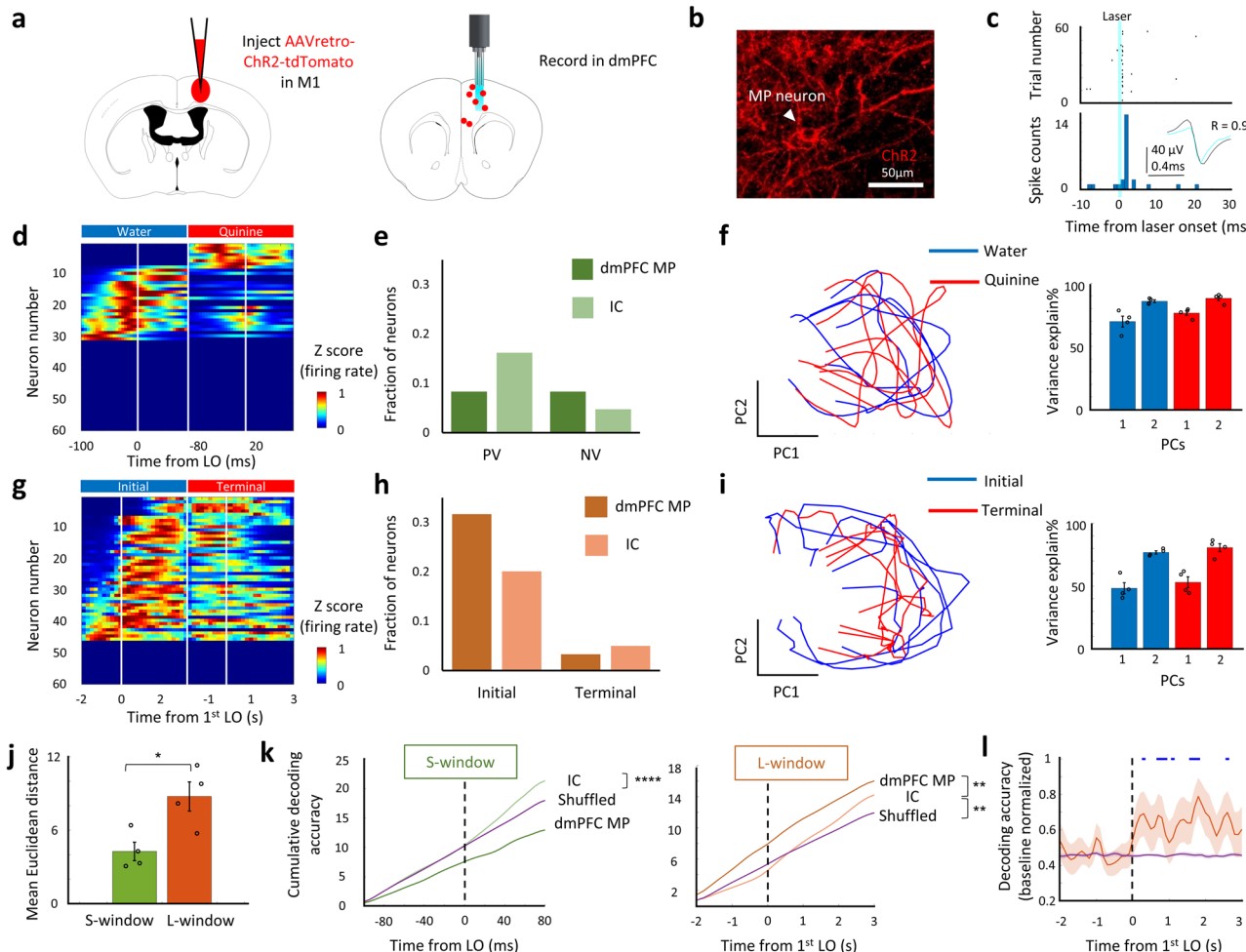

**Fig. 2 | dmPFC MP neurons represent initial phase. a** Schematics showing the labeling and recording of MP neuron in the dmPFC. Used with permission of Elsevier Science & Technology Journals, from The Mouse Brain in Stereotaxic Coordinates, Keith et al, edition 3, 2008; permission conveyed through Copyright Clearance Center, Inc. **b** Representative image showing channelrhodopsin-2 (ChR2) expression in MP neuron. **c**. Identification of labeled MP neuron. We identified the unit as MP neuron when there was a significant probability of evoked spikes, appeared from 0 to 5 ms after laser onset (light blue), and when there was a high correlation (R > 0.85) between evoked spikes (light blue waveform) and other spikes (black waveform). **d, g** Color-coded plot showing MP neural response in S-window (**d**) and in L-window (**g**) from one representative trial. **e, h** Fraction of valence (**e**) and movement phases (**h**) classified neural representations in MP and IC

neurons. **f, i** Left, PCA trajectories of dmPFC MP neuron. Right, bar plot showing cumulative variance explained percentage of first two PCs. $n = 4$ trials. Values are mean ± s.e.m. **j** Comparison of mean Euclidean distance from all PC dimensions of dmPFC MP neurons, between S-window and L-window. $n = 4$ trials. Values are mean ± s.e.m. Statistics: two-sided two sample t test, *$P < 0.05$. **k**. Decoding of taste signals (water or quinine) in S-window (left) phase (initial or terminal) in L-window (right). Statistics: two-sided two sample t test. **$P < 0.01$, ****$P < 0.0001$ represent significantly higher decoding accuracy than shuffled data. **l** Real time decoding of movement phases by dmPFC MP neurons. The neural activity is normalized by the baseline (LO-2s to LO). Blue bar represents significant ($p < 0.05$) higher decoding accuracy than shuffled data by Kolmogorov–Smirnov test.

chemogenetically silenced by administrating with CNO in mice, and the licking chances were tested during the persistent licking task. As expected, thirsty mice with chemogenetically silenced dmPFC MP neuron had lower chance to drink water ($p < 0.05$ compared to the saline trials, Supplementary Fig. 8c).

We next asked whether the effect of dmPFC MP neuron silencing was specific to the initiation of persistent licking or also general to other types of behavioral initiation. We took advantage that mice showed a phase of persistent running after a mild electric shock (Supplementary Fig. 8h). To test whether inactivation of dmPFC MP neurons also affected this behavior, we examined body activity after a 1 s electrical tail shock. Indeed, chemogenetic silencing of dmPFC MP neurons suppressed escaping behavior (Supplementary Fig. 8h, i), suggesting that dmPFC MP neurons are generally involved in the behavioral initiation. This result is consistent with previously reported general population of dmPFC neurons, silencing of which has been shown to delay the initiation of avoidance movements[19]. Indeed, since

the mice did not receive an electric shock (natural valence) after they started moving, the continuously running also suggests that the decision of the mice to persist with running was not due to natural valence.

Next, we hypothesized that the increase in the brain state of positive valence and tongue movement in the initial phase was the consequence, but not the causality, of the activation of dmPFC MP neurons. As such, the activities in IC and M1 should be affected by the optogenetic silencing of dmPFC MP neurons. To test this hypothesis, we measured the neural activity in these two brain regions with or without shining laser on dmPFC. As expected, the neuronal activities of M1 and IC were decreased ($p < 0.05$) after dmPFC MP neurons were optogenetically silenced (Fig. 4f, g). We further confirmed this initial phase specificity by excluding the effect of silencing of dmPFC MP neuron on valence at the middle phase (no significant difference of water or sucrose-licking frequency between sham and laser trials, Fig. 3f–i). Overall, our results suggest that the dmPFC MP neuron is

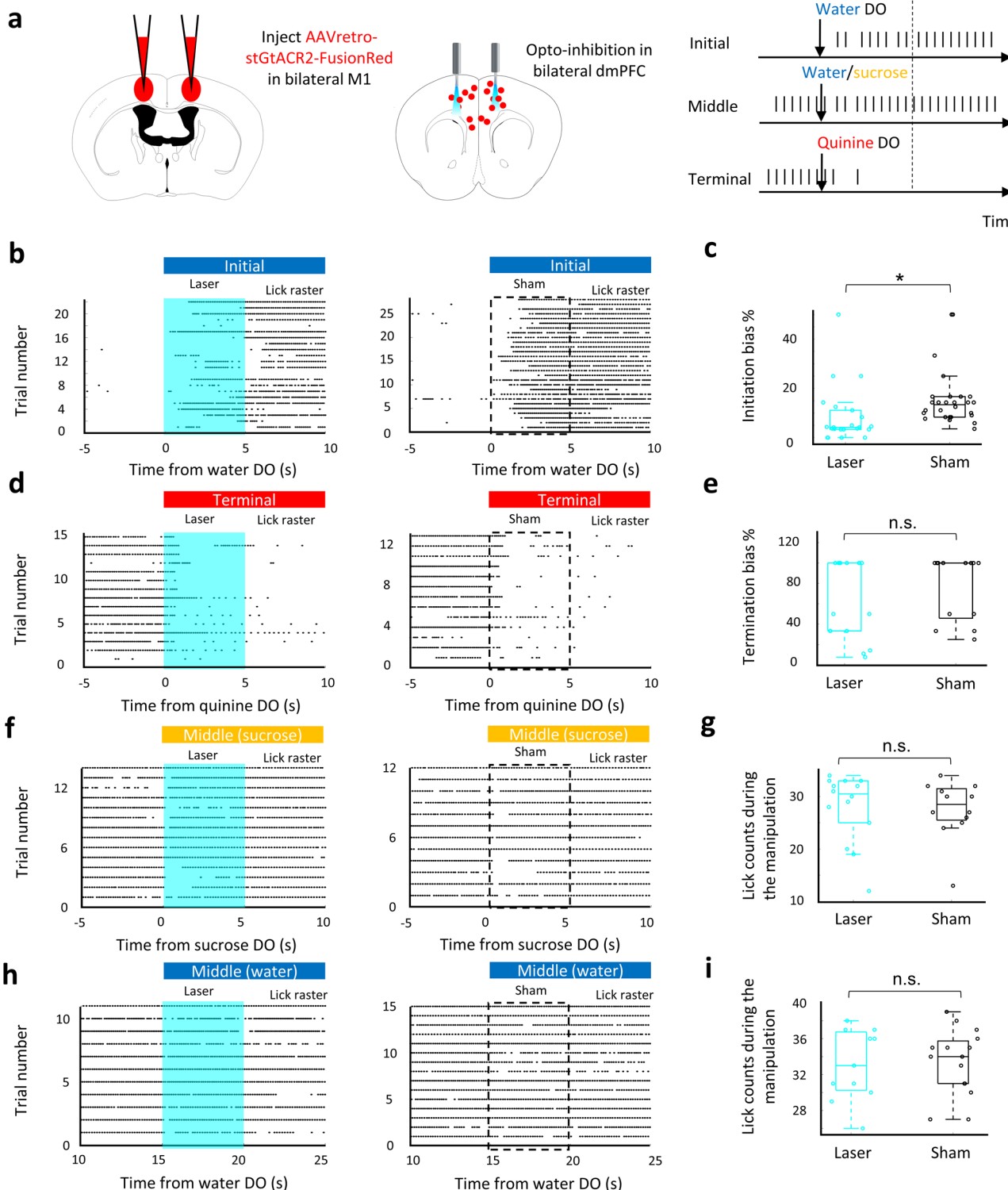

**Fig. 3 | dmPFC MP neuron is required for the initiation of persistent licking movement. a** Left two columns: schematic of bilateral silencing MP neurons in dmPFC. Used with permission of Elsevier Science & Technology Journals, from The Mouse Brain in Stereotaxic Coordinates, Keith et al, edition 3, 2008; permission conveyed through Copyright Clearance Center, Inc. Right: experimental design for optogenetic silencing. Mice received laser or sham stimulation (5 s, 20 Hz) concurrent with water or sucrose or quinine DO or water DO + 15 s. **b, d, f, h** Raster plot showing licking movement relative to water DO (b) or quinine DO (d) or sucrose DO (f) or water DO + 15 s (h). Laser (left) or sham (right) was triggered by DO. Cyan background and dash box represent laser (**b**) and sham (**d**) delivery period, respectively. **c, e.** Percentage of bias that started (**c**) or stopped (**e**) persistent lick (see Methods for the calculation of bias). $n = 22$ for the laser trials at initial phase;

$n = 28$ for the sham trials at initial phase; $n = 15$ for the laser trials at terminal phase; $n = 13$ for the sham trials at terminal phase. **g, i.** lick counts during the laser (cyan) and sham (black) manipulation. $n = 14$ for the laser trials at middle (sucrose) phase; $n = 12$ for the sham trials at middle (sucrose) phase; $n = 11$ for the laser trials at middle (water) phase; $n = 15$ for the sham trials at middle (water) phase. For all boxplot, the minima, maxima, and center bounds of box denote 25 percentile, median, and 75 percentile of data, respectively. The upper bound of whisker denote the highest data point, which lower than the sum of maxima bound and 1.5 times of box length. The lower bound of whisker denote the lowest data point, which higher than the subtraction of minima bound and 1.5 times of box length. Statistics in all panels: two-sided two sample t test, *$p < 0.05$, n.s. $p > 0.05$.

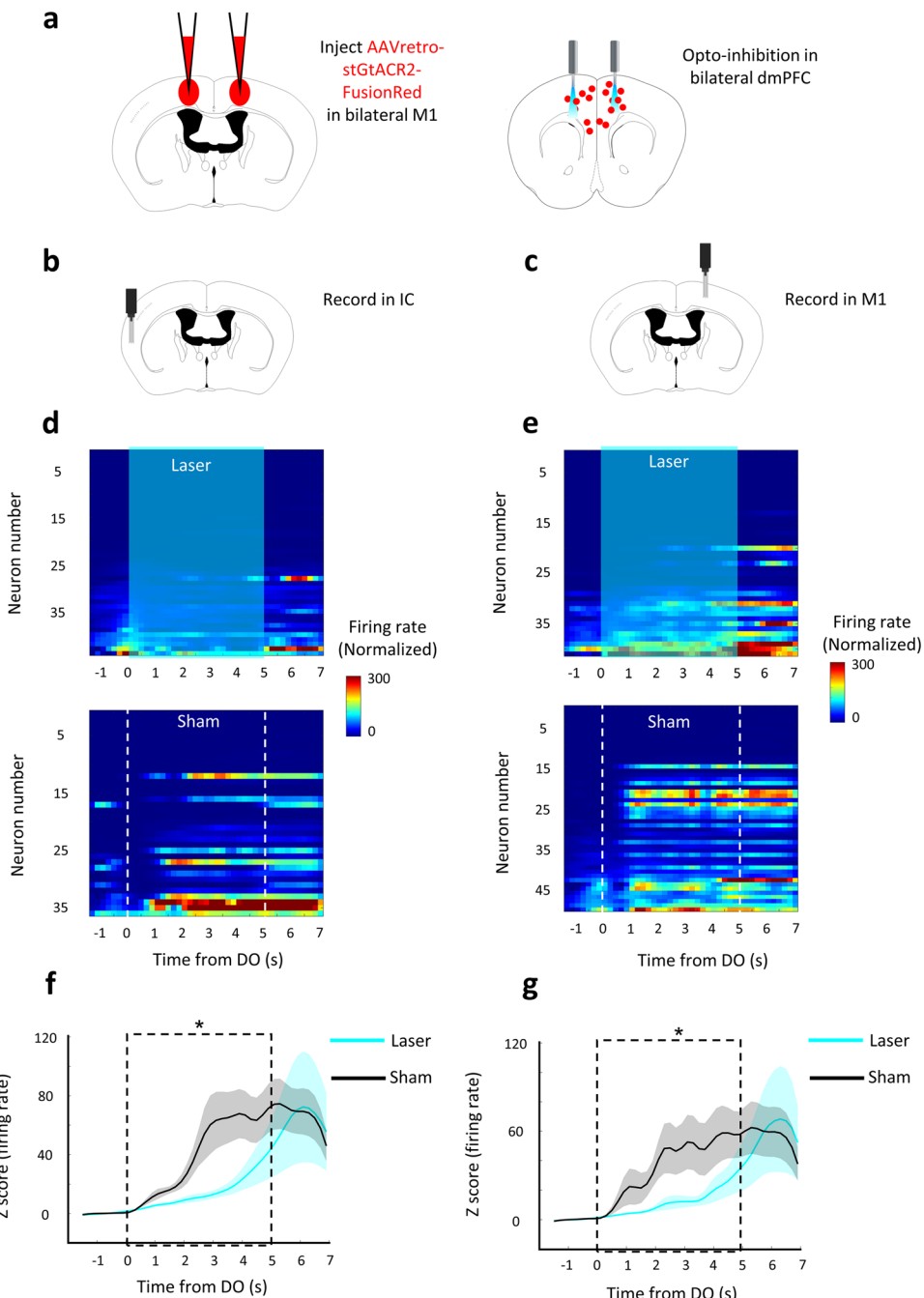

**Fig. 4 | Silencing of dmPFC MP neuron impairs the neural activity in motor and insular cortex. a** Schematic of bilateral silencing MP neurons in the dmPFC. **b, c.** Schematic of recording sites in IC (**b**) and M1 (**c**). **a–c** used with permission of Elsevier Science & Technology Journals, from The Mouse Brain in Stereotaxic Coordinates, Keith et al, edition 3, 2008; permission conveyed through Copyright Clearance Center, Inc. **d, e** Baseline subtracted, z scored firing rate, relative to water

DO, for the recorded neurons in IC (**d**) and in M1 (**e**) from one representative trial. Cyan background and dash box represent laser (top) and sham (bottom) delivery period, respectively. **f, g** Mean baseline subtracted, z scored firing rate, relative to water DO, of IC (**f**) and M1 (**g**) neurons. Values are mean ± s.e.m. Statistics: two-sided two sample t test, *$p < 0.05$.

required to initiate continuous licking, and further promoting taste valence during the initial phase, but loses its necessity in the following movement phases.

**A MP network-based computational model**

Finally, we asked what causes the MP network to initiate a persistent movement. To answer this question, we built a neural network-based model (Fig. 5a) and examined how the output of licking performance

changes in response to different types of inputs. The design of this model was mainly based on two criteria: (1) the inter-spike interval of a single neuron in the model is matched to the MP neuron in mPFC (Supplementary Fig. 10a); (2) the neuronal population of modeled network performs rotational dynamics[20] because we assumed that tongue movement follows a rhythmic pattern (Supplementary Fig. 10b, c). To verify if the output of this model matched the performance of thirsty mice in the actual experiment, we manipulated

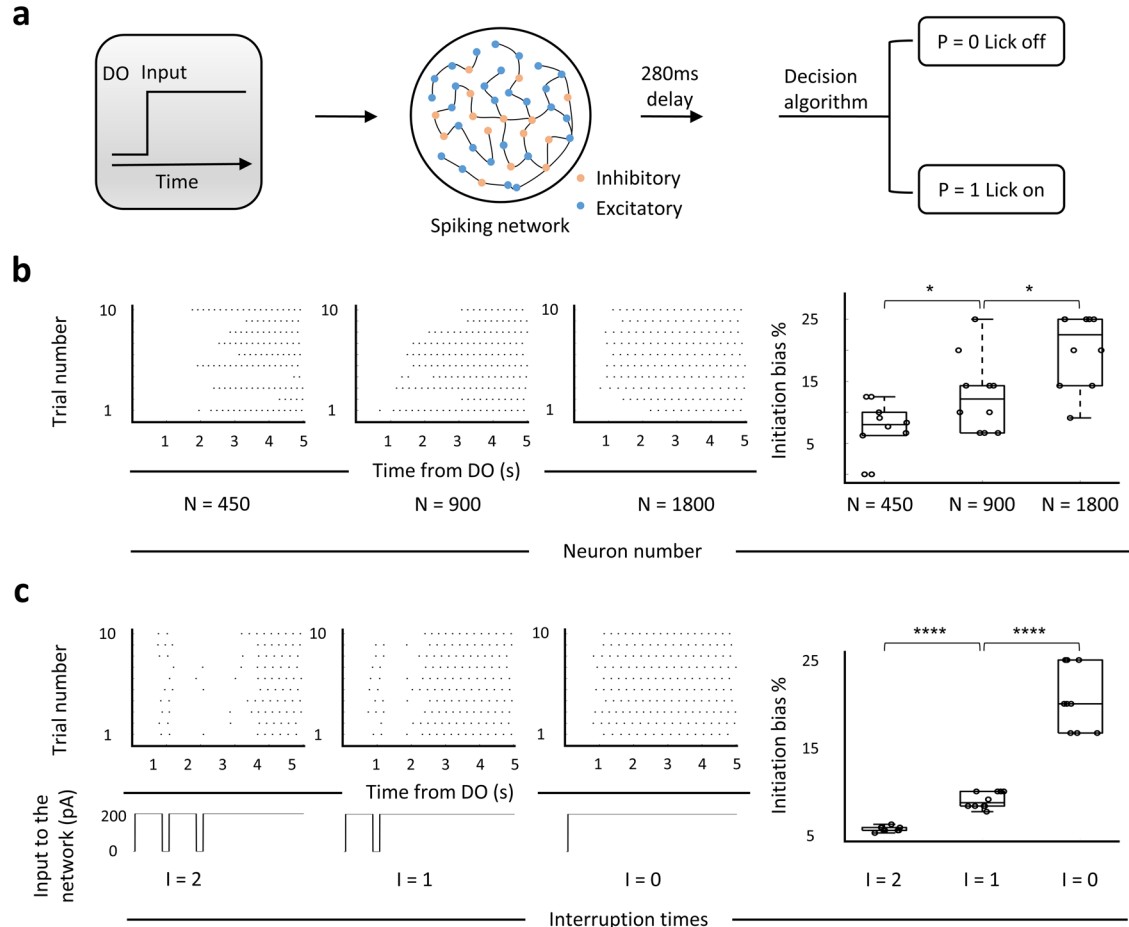

**Fig. 5 | Modulation of neuron number and input in the MP network-based computational model. a** The MP network-based model. A spiking network receives an input of current after the DO (delivery onset) with a flexible (1 to 200 ms, for panel **b**) or fixed (50 ms, for panel C) delay. 56% connectivity is excitatory and 44% of it is inhibitory. The lick raster outputs are calculated by a decision algorithm (see Methods) using the simulated spike data. **b** Simulation of optogenetic inhibition of dmPFC MP neurons by reducing the neuron number (both excitatory and inhibitory neurons) in the MP network based model. Left 1-3: lick raster produced by the model under indicated neuron number. Right: percentage of initiation bias that calculated from left lick raster data. n = 10 trials for each simulation. **c** Performance of licking behavior under the different continuities of input current. Left 1-3: top: lick raster plots under, bottom: inputs with indicated interruption times. Right: percentages of initiation bias with indicated interruption times. n = 10 trials for each simulation. For all boxplot, the minima, maxima, and center bounds of box denote 25 percentile, median, and 75 percentile of data, respectively. The upper bound of whisker denote the highest data point, which lower than the sum of maxima bound and 1.5 times of box length. The lower bound of whisker denote the lowest data point, which higher than the subtraction of minima bound and 1.5 times of box length. Statistics: two-sided two sample t test, *p < 0.05, ****p < 0.0001.

the firing rate of the simulated network by inserting the inputs with different amplitudes and examined the output licking frequency and initiation bias. We found no linear relationship between the mean neuronal firing rate and the above two parameters (Supplementary Fig. 10d, red), which is consistent with the experimental observations (Supplementary Fig. 10d, black). To further ensure the feasibility of the MP network-based model, we simulated the optogenetic silencing of dmPFC MP neurons through decreasing the number of all neurons in the modeled network. Consistent with the experimental data, the network with reduced neural population inhibited the initiation of persistent lick (Fig. 5b). We then examined the effect of temporally continuous input on the output of the network. We found that even a single short-term interruption (200 ms) disrupted the initiation of continuous lick (average 57% decrease on initiation bias, Fig. 5c). The percentage of bias with two interruptions decreased to 5.7 ± 0.08% (compared with 21 ± 1.17% with no interruption, Fig. 5c). This temporally continuous input of the model suggests that the triggering signal for the MP network should be intact, continuous sensory stimulation. For the persistent licking task, we suspected this sensory stimulation may come from the continuous sensation of combined liquid delivery and internal thirst.

## Discussion

Our study showed that after receiving a sensory signal, dmPFC MP neurons can send command signals to the primary motor cortex and striatum, which in turn initiate the downstream machineries for a persistent action (Fig. 6). Silencing of dmPFC MP neurons disrupts the association between sensory signals and motor command. As a result, the initiation of persistent movement is impaired. Our results suggest that the decision in the mPFC whether to persist with the current action is mainly based on the contextual information in the absence of opposing valence in initial phase of a persistent movement. Based on our computational model, we reasoned that contextual information such as thirst or the sensation of water delivery can provide a continuous signal, whereas natural valence is discrete, so contextual information can trigger a more efficient and continuous movement. However, when an opposing valence appears, other circuits may be involved to terminate the movement.

Given that most frontal neurons tune to abstract variables[21,22], we speculated that the dmPFC MP neurons are generally used for instructing the subsequent movement patterns (e.g. whether it should be discrete or persistent). Although we did not investigate the coding of dmPFC MP neurons in a discrete movement, we suspected that

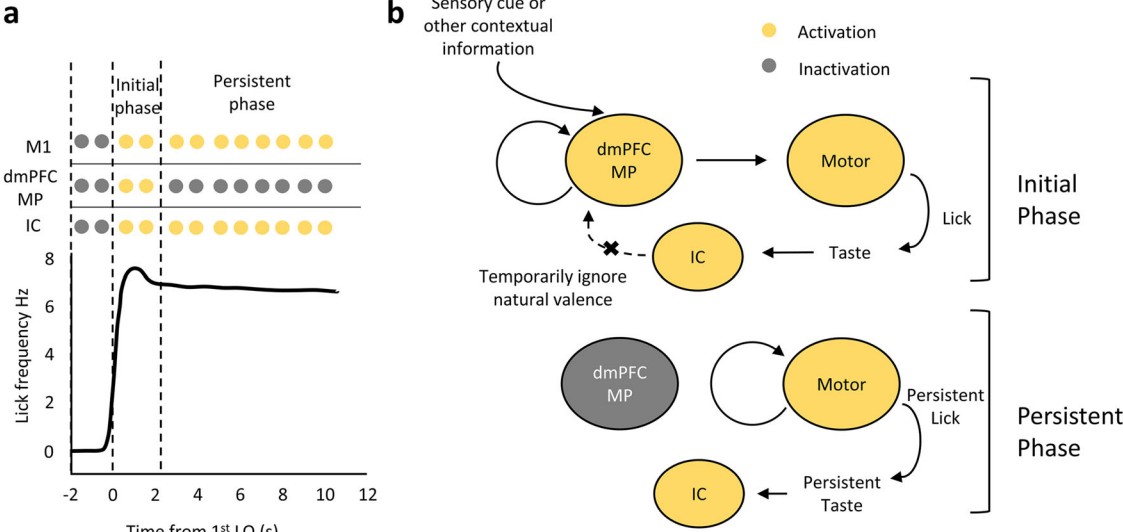

**Fig. 6 | Summarized model for how dmPFC MP neurons initiate persistent movement. a** Pattern of neuronal activation during the persistent licking movement. After the onset of stimulation, there is a delay of approximately 1.3 seconds until the first lick onset. In the initial phase (the licking frequency is driven to the maximum), the indicated three types of neurons are activated. In the persistent phase, dmPFC MP neurons are inactivated, while M1 and IC neurons show persistent activity. **b** During the initial phase, when the sensory signals are received,

dmPFC MP neurons activate themselves and the downstream motor neurons. The latter drive the licking movements that bring the tongue into contact with the water. The taste signal from the tongue is further transmitted to the valence-encoding neurons in the IC. During the persistent phase, no signal is transmitted from the dmPFC MP neurons to the motor neurons, but the motor neurons is continuously activated. This results in persistent licking movement and persistent valence encoding in the IC.

dmPFC MP may also carry valence signal to direct a discrete movement so that the licking frequency can be correlated with the valence level[8]. Different from the neurons in premotor cortex, the dmPFC MP neurons do not tune to specific latent motor variables, such as lick angle (Supplementary Fig. 9b, c, d). Since the whole premotor and prefrontal cortex can be seen as integrated premotor cortex[23], the dmPFC MP neurons may serve as a role in the low-level of a hierarchy of PFC circuits and high-level of the hierarchy of the premotor circuits.

Although IC showed stable valence coding across lick trials in a persistent licking movement (Supplementary Fig. 6a3), this coding was not persistent (the slopes of trend line deviate significantly from zero; Supplementary Fig. 4d, e). This is probably because different neurons represented the positive or negative valence in different phases. Only the neuronal representations of the positive valence in the initial phase were able to send the signal to the dmPFC. This possibility could also explain why the encoding of valence in the dmPFC faded across the lick trials in a persistent movement.

## Methods
### Subject details
All experimental procedures were approved by the Institutional Animal Care and Use Committee (IACUC) and the Biosafety Committee of the University of Wyoming. 14 male and 6 female immunocompetent mice of specified age (indicated in **Surgeries** and **Behavioral details**) were used for various experimental purposes. All mice were bred on a C57/BL6J background. Mice older than 30 days were housed with same-sex littermates or alone in a vivarium at 21-23 ºC with 25%-30% humidity and a 12-h light/dark cycle. Mice implanted with an electrode or head bar were housed alone. For chemo-genetic experiments, 2 male and 2 female mice received AAV injection and head bar implant. For optogenetic experiments, 6 male mice and 2 female mice received AAV injection, an opto-electrode, and head bar implantation. 3 male mice and 1 female mouse were implanted with electrodes and head bar. 1 male mouse and 1 female mouse were implanted with optical fibers and electrodes. Optical fibers were implanted in 2 male mice.

### Surgeries
The preparatory procedures are similar for both implantation and injection. Mice were anesthetized using oxygenated (2 LPM for induction and 0.4 LPM for maintenance) 2% isoflurane (v/v). Mice heads were fixed to the stereotactic device (NARISHIGE SG −4N) and maintained at 37 ºC with a heating pad (K&H No. 1060). Seventy percent isopropyl alcohol and iodine were placed on the incision site. The skull was exposed by cutting the skin and removing the dura and connective tissue. The coordinates used for positioning the injection and implantation sites were relative to Bregma (anterior-posterior A-P, medio-lateral M-L, dorsal-ventral D-V) in mm. After the surgeries, the mice were administered intraperitoneal ibuprofen (50 mg kg⁻¹), and they were kept at 37 ºC for 30-60 minutes before returning to the home cage.

For viral injection, P14-30 mice were used. A small craniotomy (approximately 0.2 mm in diameter) was made over the injection site. Glass filaments (Drummond Scientific Co.) with a tip diameter of approximately 5 μm were filled with 2 μL of virus solution. By pressure injection with a custom-made device driven by a single-axis hydraulic manipulator (NARISHIGE mmo-220A), the viral solution (undiluted, 100 nL at each injection site) was delivered to the desired regions at a rate of 30-50 nL min⁻¹. Opto-labeled dmPFC MP neurons were labeled with pAAV- CAG -hChR2-mcherry[24] (Addgene viral prep # 28017-AAVrg) injected into motor cortex (A-P −0.6, M-L 1.0, D-V 0.2 0.5 0.8). For chemogenetic inhibition, pAAV-Ef1a-mCherry- IRES -Cre[25] (Addgene viral prep # 55632-AAVrg) was injected into bilateral motor cortex (A-P −0.6, M-L ± 1.0, D-V 0.2 0.5 0.8), followed by injection of pAAV-hsyn-DIO-hm4D(Gi)[26] (Addgene viral prep # 44362-AAV5) into the bilateral dmPFC (A-P 1.35, M-L ± 0.2, D-V 0.2 0.5 0.8 at an angle of 30° to the upright position). For optogenetic silencing, pAAV-CKIIa-stGtACR2-FusionRed[16] (Addgene viral prep # 105669-AAVrg) was injected into the bilateral motor cortex (A-P −0.6, M-L ± 1.0, D-V 0.2 0.5 0.8). Mice were returned to the home cage until at least three weeks before implantation.

For implantation, silicon probes (A4x8-Edge-2mm-100-200-177-CM32 or A1x32-Edge-5mm-25-177-CM32, NeuroNexus) or 32-tetrode

bundles (Bio-Signal technologies) were implanted followed by optic fibers and head bar. To build opto-electrode, optic fibers (MFC_200/245-0.37_2.0mm_MF1.25_FLT) were fixed around 0.5 mm above the electrodes using crazy glue and dental cement (Lang Dental). In opto-inhibition experiments, two optic fibers (MFC_600/710-0.37_1.0mm_MF1.25_FLT) were implanted to the bilateral prefrontal cortices (A-P 1.7, M-L ± 0.5, D-V 0.5). To fit the shape of prefrontal, motor, and insular cortex, customized 32-tetrode bundles were split into one or two clusters. To record the single unit, electrodes were implanted in the left hemisphere with the following designs and coordinates: silicon probe (A1x32-Edge −5mm-25-177-CM32) was implanted at the pIC (A-P − 0.5-(−1.5), M-L 3-4, D-V 3); silicon probe (A1x32-5mm-25-177-CM32) was implanted at the aIC (A-P 1.5-1.7, M-L 2.5-3.5, D-V 2.5); silicon probe (A4x8-Edge-2mm-100-200-177-CM32) based opto-electrode was implanted at the dmPFC (A-P 1.0-1.7, M-L 0.1-1.5, D-V 1); 32-tetrode bundles (one-cluster) based opto-electrode was implanted at the dmPFC (A-P 1.0-1.7, M-L 0.1-1.5, D-V 1); 32-tetrode bundles (two-cluster) were implanted at the pIC (A-P − 0.5-(−1.5), M-L 3-4, D-V 3) and M1 (A-P 1-2, M-L 1-2, D-V 0.5). These coordinates were designed based on the Mouse Brain Stereotaxic Coordinate[27] (MBSC). To distinguish the M1 from dmPFC and reduce the interference from the dmPFC, the coordinate of M1 deviated from jaw/tongue motor cortex[17]. However, we confirmed that lick specific signal can still be collected from this region (Supplementary Fig. 4a1, b1, c1). It is probably because jaw/tongue cortex controls lick direction[17] while more posterior motor cortex is correlated with tongue in and out. During the surgery, the skull was horizontally aligned through a fixing apparatus (Stoelting Co.). An anchor screw was placed on the right cerebellum to connect ground wires of the electrodes. After placing the anchor screw and electrodes, silicone sealant (kwik-cast, world precision instrument) was applied above the exposed brain tissue. A customized head bar (github.com/ywang2822/Multi_Lick_ports_behavioral_setup) was then positioned over the skull. To affix the implant, Metabond (C&B Metabond, Parkell) and dental cement (Lang Dental) were applied. The behavioral experiments started at least one week after the surgery.

## Behavioral details

The head-fix setup was connected to a construction rod (Throlabs) by a 3d printed connector (github.com/ywang2822/Multi_Lick_ports_behavioral_setup). Multi-lick-ports (github.com/ywang2822/Multi_Lick_ports_behavioral_setup) were placed in front of the head fix and connected to the Dual Lick Port Detector (www.janelia.org/open-science/dual-lick-port-detector). Three Clearlink sets (Baxter) were used for liquid delivery. The delivery speed was manually calibrated to 0.15–0.2 mL min⁻¹ every time before the behavioral test. The delivery switch was controlled by three solenoid valves (LFVA1220210H, THE LEE CO.) in a noise-reducing box. The switch timing was programmed through the Bpod (Sanworks). The signal of mice locomotor activity was collected through an optical shaft encoder (H5-360-IE-S, US digital). For facial videography set-up, the camera (S3-U3-91S6C-C, Teledyne FLIR) was positioned at the right side of the mouse's lateral face surface, which illuminated by two infrared arrays. For laser delivery, a solid-state laser (Shanghai Laser& Optics Century Co., 473 nm) was connected to fiber optic patch cord (Doric Lenses), which attached to the implanted optic fibers using ceramic mating sleeves. To conditionally control the laser delivery by water, sucrose, or quinine onset, we used a 4-way data switch box (BNC, Kentek) to bridge the laser and solenoid valves. A programmable stimulator (A-M system, model 4100) was used to control laser delivery and a voltage pulse for tail shock experiment. All signals, including frame timing, wheel speed, liquid delivery timing, lick timing, shock timing, and laser delivery timing, were sent to an USB interface board (Intan Technologies, RHD).

For licking task, to induce persistent behavior in mice while keep their health as much as possible, we used an acute water deprivation protocol. In our protocol, mice were deprived of water for 16 to

36 hours until their body weight decreased by approximately 22%. After the experiment, the mice were returned to the home cage where they had unlimited access to water for at least five days or until their body weights fully recovered. We repeated this procedure for 2 to 3 months and continuously monitored the well-being of the mice. According to our observations, the body weight of the mice increased in the long term after we started this protocol (average 8.31% ± 1.9% at 60th day, Supplementary Fig. 11a). Similar weight gain was also observed in normal and mild water restricted laboratory rodent[28]. Mice did not perform significantly decreased locomotor activity after water deprivation (Supplementary Fig. 11b), which suggests that mice were not in distress. This result is different from one-time acute water deprivation, which is caused apparent distress when excess 24 hours[29]. We reasoned that mice can adapt to regular acute water deprivation. Although water deprivation longer than 24 hours is not recommended[29], this time limit largely depends on the individual conditions[30]. Indeed, water deprivation time is significantly various from mouse to mouse (Supplementary Fig. 11c) to acquire approximately 22% weight loss. Besides body weight, the water deprivation time may also depend on the body water percentage, calorie consumption, nocturnal/diurnal deprivation time ratio, et al, because there is no significant linear correlation between body weight and water deprivation time (Supplementary Fig. 11d). Therefore, the 24-hour time limit is not fixed. As for the percentage of body weight loss, weight loss greater than 15% is also not recommended[29]. However, it may also depend on whether the mice are deprived of water once or several times. Based on a widely used deprivation protocol, mice can remain healthy for four months even after their body weight has stabilized at about 80% of body weight[31]. It suggests that mice can adapt to a new stressful environment.

During training phase, mice learned to sense the water drop through their whiskers or jaws. We considered mice to become proficient at the task when licking happened within 3 s after the delivery onset (DO) in all repeated trials. During the test phase, we first delivered water and 20% sucrose in a random sequence for a total of 30 s. After at least 5 min, we then orderly delivered water, 20% sucrose, and 5 mM quinine for 10 s each or water and 5 mM quinine for 15 s each.

For tail shocking task, 16-23 volts electrical shocks were administered to the tail by a customized shocker (electric shock box machine kit, STEREN). Two conductive adhesive copper tapes were connected to the shocker and positioned 2 cm apart at the tail by sticking on customized heat shrink tube (various on the circumference of mouse tail). During the first time of training, the voltage of electrical shocks were adjusted until escaping behavior was observed (speed>10 cm s⁻¹ right after the shock). This voltage was recorded and used for the following tests. Those who did not perform escaping behavior were excluded from the test.

## Spike sorting and firing rate estimation

Before spike sorting, single unit data were acquired from 32-channel RHD head stage, which connected with a signal acquisition system (USB board, Intan Technologies) with sampling rate at 20 kHz. All spike sorting procedures were performed with an offline software Spikesorter[32–34] (swindale.ecc.ubc.ca/home-page/software/) under following parameters: (0.5 kHz and 4 poles high pass Butterworth filter) for signal filtering, (noise calculation: median; threshold: 80μV, 5x noise, 0.75 ms window width) for even detection, and (pca dimensions = 2; template window: −0.8 to 0.8; starting sigma = 5; threshold = 9) for clustering. We used Bayesian adaptive kernel smoother[35] with following parameters, α = 4 and β = (number of spike events) ^ (4/5), to estimate the firing rate of sorted spikes. For small scale temporal window (180 ms), we used a bandwidth of 5 ms. While for large scale temporal window (5 s), we used a bandwidth of 200 ms.

## Optogenetic silencing

We illuminated bilateral prefrontal cortices using 473 nm 5 mW laser to activate stGtACR2[16]. Laser pulses (40 ms width at 20 Hz) were delivered in a 5 s duration. The onset of laser pulses was triggered based on either water DO or quinine DO. The optogenetic silencing experiments were only performed after mice reached stable behavioral level (after at least two test phases and (lick onset (LO) − DO < 3 s) in all test phases). The trials, of which licking frequency > 0.5 Hz in the time course 3 s before DO, were excluded. Histological characterizations were used to identify the viral infection.

## Opto-tagging

We applied 473 nm 7 mW laser pulses (1 ms width at 20 Hz, 3 s duration) on the unilaterally prefrontal cortex of viral (AAV-ChR2) injected mice. Laser and network-evoked spikes (see also Spike sorting and firing rate estimation) were identified using the Stimulus Associated spike Latency Test (SALT)[15]. Specifically, laser and network-evoked spikes were assessed in a 0–5 ms and a 6-10 ms temporal window after laser onset, respectively. For those units with significant correlation (correlation coefficient > 0.85) of average waveform and significantly different distribution ($P < 0.05$) of spike latency with baseline units were identified as laser or network-evoked units.

## Chemogenetic inhibition

Viral pAAV-hsyn-DIO-hm4D(Gi) (Addgene_44362-AAV5) injected mice were administered intraperitoneally with Clozapine N-oxide dihydrochloride (CNO, 2 mg kg⁻¹, Tocris) ten minutes before the licking or tail shocking task. Only the mice reached stable behavioral level (after at least two test phases and (LO − DO < 3 s) in all test phases) were used for chemogenetic experiments. In the licking task, mice were retrained to lick the water one to two times after recovery from CNO administration. The re-trained phases were not included in test phases.

## Analysis of facial and locomotor activity

We collected the frames during the licking or tail shocking task. We then converted these frames into histogram of oriented gradients (HOG) vectors by using 8 orientations, 32 pixels per cell and 1 cell per block. To extract the most variant facial part, we cropped the ear part with 364×296 pixels fixed size and manually selected position of each transformed HOG vector[36]. Temporally adjacent HOG vectors were paired, the facial activity at each time point was calculated as follows: $1-\Delta R$, where $\Delta R$ is the correlation coefficient between two temporally adjacent HOG vectors.

The signals that collected from the encoder were digital pulses. The locomotor activity was calculated as speed (cm s⁻¹): $\frac{circumf}{CPR \cdot dt}$, where $circumf$ is the circumference (cm) of the wheel, $CPR$ (cycles per revolution) is 360, and $dt$ is the time interval between two digital pulses.

## Analysis of licking initiation/termination bias

With the feeling of extremely thirsty, the mice will start a non-stop licking behavior when water is available until feeling satiated or the delivery stopped[37] (Supplementary Fig. 1d). To evaluate if the mice start or stop the continuous, but not discrete, lick, we calculated the initiation and termination bias. We first calculated simple moving averages (SMAs) after water or quinine DO as following: $SMA = \frac{\sum_{i=1}^{n} l_i}{n}$, where $l$ is the lick times during a 200 ms time window and $n = 5$. For the initiation bias, all values of SMAs were ignored if there was a zero value after DO. We created a vector that contained SMAs sampled at 200 ms intervals. The SMA value was counted from the last non-zero value. The initiation bias (ibias) was calculated as: $ibias = \frac{1}{idx}$, where $idx$ is the first time point of the SMA > 1.2 (6 Hz). If all SMA values equal zeros in 6 seconds, $ibias$ was set as zero. For the termination bias, the SMA value was counted from the first time point after the quinine DO (for the water-quinine and water-sucrose-quinine session) or the end time

point of water delivery (for the session water-water session) or sucrose delivery (for the session water-sucrose session). The termination bias (tbias) was calculated as: $tbias = \frac{1}{idx}$, where $idx$ is the first time point of the SMA < 1 (5 Hz). If all SMA values ≥ 1, $tbias$ was set as zero.

## Cell classification

For the cell classification to discriminate water- and quinine-licks, we categorized single-units into four separate groups of neural representation (lick, positive valence (PV), negative valence (NV), and mixed valence (MV)) based on firing rate estimation at the lick window (LO-100ms: LO + 80 ms). To determine if the firing rate is significantly higher than normal condition, we created pseudo-trials that have the same lick interval with the corresponding real licks during the 10 s baseline. Individual time bins of each pseudo lick trials were concatenated horizontally and shuffled. This procedure was repeated 1000 times and the pseudo-trial matrix was calculated as the mean among shuffled datasets. For real lick trials, we only selected the first four trials for encoding analysis. At each individual time bin of pseudo and real trials, we calculated Euclidean norm of two temporally adjacent firing rate estimations (5 ms each). When absolute z-scores of two distributions ($z_{12} = \frac{\mu_1 - \mu_2}{\sqrt{\sigma_1^2 + \sigma_2^2}}$, where $\mu$ and $\sigma$ represent mean and standard deviation, respectively, of the distribution1 and 2) exceeded 1.29, they were considered significantly different. We selected single-units with significantly high firing rate in water or quinine lick trials for further analysis. To evaluate the time bias of firing rate across lick trials, we mean centered the whole firing rate matrix. The Frobenius norms were calculated as follows: $norm = \sqrt{\sum_{i=1}^{n}(t_i - c)^2}$, where $t_i$ is the mean column value of firing rate matrix with 10 ms time bin across lick trials, $c$ is centered mean, and n equals the number of time bins. We categorized a single-unit with the time bias of lick trials when its real norm greater than 95% of 1000 shuffled norms. To estimate if the peak of firing rate of two distributions is different, we compared the times of the maximum firing rate across lick trials between two distributions using two sample t-test. Their firing rate peaks were considered different when p-value less than 0.05 (z-score>1.64). Single-units were categorized into the group of lick, positive or negative valence, or mixed response when their firing rates met following conditions: lick, quinine trials > pseudo trials & water trials > pseudo trials & without time bias of the firing peak between quinine trials and water trials & with the time bias of water and quinine trials; positive valence (PV), water trials > pseudo trials & water trials > quinine trials & quinine trials ≤ pseudo trials; negative valence (NV), quinine trials > pseudo trials & quinine trials > water trials & water trials ≤ pseudo trials; mixed valence (MV), water trials > pseudo trials & quinine trials > pseudo trials & with time bias of the firing peak between quinine trials and water trials & with the time bias of water and quinine trials; others were grouped into unrelated valence (UV).

To examine if there is a trial bias of firing rate during the water-licks, we compared the spike times at the small scale time window of water-licks. The matrices were binned (8 trials per group) and the mean values of each group were calculated. New generated binned firing rate matrices were then used. We next calculated Frobenius norm: $\sqrt{\sum_{i=1}^{n}(t_i - c)^2}$, where $t_i$ is the mean row value of the binned firing rate matrix, $c$ is centered mean, while $n$ is the number of trials. When the real Frobenius norm was greater than 95% of 1000 shuffled norm, the single-unit was considered with trial bias in water-lick trials.

For the cell classification to discriminate initial and terminal phase of persistent movement, we assessed the firing rates at the 2 s temporal window before or after the first water LO or quinine DO. The Five time point (0 s, 0.5 s, 1 s, 1.5 s, and 2 s) were used for classification analysis. To construct pseudo data, the over 70 s spike train data were

used to extract five-time point random temporal window. We defined a single-unit with estimated firing rate higher than 65% pseudo data at the temporal window (1st water LO-2s: 1st water LO) and lower than 35% pseudo data at the temporal window (quinine DO: quinine DO + 2 s) as initial phase neural representation before LO; with estimated firing rate higher than 65% pseudo data at the temporal window (1st water LO: 1st water LO + 2 s) and lower than 35% pseudo data at the temporal window (quinine DO: quinine DO + 2 s) as initial phase neural representation after LO; with estimated firing rate higher than 65% pseudo data at the temporal window (quinine DO: quinine DO + 2 s) and lower than 35% pseudo data at the temporal window (1st water LO-2s: 1st water LO + 2 s) as terminal phase neural representation. Others were classified as neural representations of unrelated movement phases.

Since the two classification methods mentioned above have to be aligned, the cell classifications were performed in a single trial. Therefore, these two cell classification methods cannot be used to define functional cell types. They were only used to assess whether valence and movement phases coding are mixed or not.

### Connectivity estimation
To estimate the connections among different neural representations, we used Total Spiking Probability Edges (TSPE[14]). This method allows us to calculate the cross-correlation between pairs of spike trains and to evaluate excitatory connections. Furthermore, it gives high accuracy estimation in a short recording period. However, this method can only be used for the comparison of two individual single-units. To apply this method on two networks, we first selected spike trains of photo-tagged and network evoked single-units. We assumed the connection from photo-tagged to network evoked single-units is positive. For the individual single-unit, 3 s time duration of spike times data was cropped and sent to TSPE calculation. We next selected spike trains of the neural representations of NV, PV, and initial phase in three brain regions (IC, M1, and dmPFC). Individual time windows of 180 ms across the lick trials (LO-100ms: LO + 80 ms) were extracted and recombined. The connectivity among these neural representations was calculated using TSPE. To perform a statistical test, we compared the real TSPE and the pseudo TSPE, in which 3 s spike times data were shuffled 200 times and created pseudo spike times. 70, 80, 90, 95, and 99 percentile of pseudo TSPE were used for the comparison based on the neural representation number of real data. Specifically, if the real data number is less than 10, they will not be used for the comparison; if the real data number is between 10 and 15, 80 percentile of pseudo TSPE will be used for the comparison; if the real data number is between 15 and 20, 90 percentile of pseudo TSPE will be used for the comparison; if the real data number is between 20 and 32, 95 percentile of pseudo TSPE will be used for the comparison; if the real data number is larger than 32, 99 percentile of pseudo TSPE will be used for the comparison.

### Decoding analysis
For both small and large scale temporal window decoding, we employed a multiclass linear discriminant analysis. We first estimated spike firing rate (see **Spike sorting and firing rate estimation)** in water and quinine lick window (LO-100ms to LO + 80 ms for small scale window and first LO-2s to first LO + 3 s for large scale window). The firing rate was normalized by dividing the maximum firing rate value of water and quinine lick window. To create firing rate pseudo data, pseudo LOs were randomly selected for the entire time duration of the spike train. This procedure was repeated 50 times. Same with real data normalization, the pseudo data were divided by the max firing rate value in each repeat. To improve the subsequent decoding performance, we reduced the dimensionality of the neural activity by PCA. The dimensions that explained over 85% of the data variance were selected to train a decoder, which is based on an error-correcting output codes (ECOC) classifier using binary support vector machine (SVM) learner (MATLAB 'fitcecoc' function). 50 percent of real and pseudo data were used to train the decoder, and the rest of them were used to test the decoder's performance. This procedure was repeated 10 times to get an average accuracy and standard deviation.

### Facial activity prediction
We trained Hammerstein-Wiener model to predict behavioral activity using estimated firing rate[38]. Before the modeling, the firing rates were estimated in a large scale temporal window (first LO-2s to first LO + 3 s) and normalized into a range from 0 to 1 by dividing the maximum firing rate value of water and quinine lick window. Empty and zero firing rates vectors were removed. The facial dynamics (see Facial activity analysis) were smoothed using Gaussian filter. The parameters that can best simulate facial dynamics were used to construct Hammerstein-Wiener model. Specifically, the number of zeros was set in a range from 0 to 2 (nb-1); the number of poles was set in a range from 1 to 3 (nf); and the degree of input nonlinearity estimator (one-dimensional polynomial) was set from 2 to 5. We then used MATLAB function 'predict' to obtain decoding accuracies of facial dynamics from the test spike data. Specifically, the data (four trials in total) were split 50/50. First two trials were used for training the model and last two trials were used for testing. To avoid and overflow error, the values which lower than negative 1500 were excluded.

### Histology
To check the position of implanted electrodes and site of injection, mice were anesthetized with 2% isoflurane (v/v) and perfused intra-cardially with 0.9% saline followed by 4% paraformaldehyde (PFA). Fixed brains were washed three times before dehydration in 30% sucrose for 24 hr. Slices were cut on a cryostat (MICROM, HM505E) at 70μm thickness after embedding with an optimal cutting temperature compound (Tissue tek). Fluorescent images were acquired by an LSM 980 microscope (Zeiss), with a 10 × 0.45NA objective or a 2.5 × 0.085NA objective.

### MP network-based model
The purpose of this modeling was to create lick raster readouts through giving a type of current input, simulating a network of neural activities, and transforming the simulated spike timing to the lick timing.

For the input current, we set the starting time $t_s$ with a flexible (1-200 ms) or a fixed (50 ms) delay after the DO. The amplitudes of input current were varied from 50-200pA according to the different simulations. To test the effect of input continuity, the current was cut using one or twice 200 ms zero amplitude.

We simulated neural activity using Brian2 simulator package in a customized python code based on a sparsely connected spiking neuron network[39]. The network consisted of 1000 excitatory and 800 inhibitory neurons as default (we decreased this number in the simulation of optogenetic inhibition). The membrane potential of each neuron was modeled according to the MP neuron membrane properties[4] based on the Hodgkin-Huxley model as following:

$$\frac{dV}{dt} = \frac{gl \cdot (El - V) - g_{Na} \cdot m^3 \cdot h \cdot (V - E_{Na}) - g_K \cdot n^4 \cdot (V - E_K) + g_e \cdot (E_e - V) + g_i \cdot (E_i - V) + I_{ex} \cdot t_s}{C_m} \tag{1}$$

$$\frac{dg_e}{dt} = -\frac{g_e}{\tau_e} \tag{2}$$

$$\frac{dg_i}{dt} = -\frac{g_i}{\tau_i} \tag{3}$$

where the excitatory and inhibitory synaptic time constants $\tau_e$ and $\tau_i$ were set as 5 and 10 ms, respectively. Other metrics were adjusted to

adapt the inter-spike-intervals of L5 mPFC MP neuron, which proposed to be a functional and dominate interneurons that bridge the gap between deep brain regions and motor cortex[4]. The membrane potential $V$ was initiated randomly at a range from −65mV to −63 mV. The excitatory conductance was set at a range from 0 to 0.06 ns and the inhibitory counterpart was in 0 to 1.5 ns.

The synapse action was dependent on the usage $u$ and availability $x$ of released neurotransmitter before and after an action potential as following: $\frac{du}{dt} = -\omega_f \cdot u$, $\frac{dx}{dt} = \omega_d - \omega_d \cdot x$, where the facilitation rate was set as 3.33 s⁻¹ and the depression rate was set as 2 s⁻¹. The rest synaptic release probability was set as 0.6. The probability of excitatory connection of the network was given as 0.1 and the inhibitory counterpart was 0.2.

To generate lick raster data from the network, we first assumed that one single lick cycle is governed by a rotational neural dynamic[20]. Then we divided one cycle of the lick into nine phases. The triggering probability $P$ of a lick signal was calculated from the network through a decision algorithm:

$$P(t) = \begin{cases} 1, & \prod_{\theta=0}^{2\pi}\left(\sum_{i=1}^{N}\varphi_{\theta i}(t-280)\right) > 0 \, \& \, S\left(\sum_{i=1}^{N}\varphi_{\theta i}(t-280)\right) < 30 \\ 0, & \prod_{\theta=0}^{2\pi}\left(\sum_{i=1}^{N}\varphi_{\theta i}(t-280)\right) = 0 \, or \, S\left(\sum_{i=1}^{N}\varphi_{\theta i}(t-280)\right) \geq 30 \end{cases}$$
(4)

where $S$ represents the standard deviation of the spike counts in whole phases.

### Statistical test

For comparison of the mean of facial and locomotor activity (Supplementary Fig. 1k-n; Supplementary Fig. 8e, h; Supplementary Fig. 9 a1 & b1 & c1 & d1), comparison of the mean of licking frequency at different sessions (Supplementary Fig. 1c–f), and comparison of TSPE (Supplementary Fig. 5), we used two-tailed Wilcoxon signed rank test. For evaluation of the slope of spike counts trend line, we used one sample t test (Supplementary Fig. 4e). Kolmogorov-Smirnov test was used for comparing normalized decoding accuracy between real and shuffled data (Fig. 2l). The rest of statistical tests were two sample t test.

### Reporting summary

Further information on research design is available in the Nature Portfolio Reporting Summary linked to this article.

## Data availability

The raw data that support these findings are available from corresponding author upon request. Source data are provided with this paper.

## Code availability

Materials and setup instruction for head fix and triple lick ports are freely available (https://github.com/ywang2822/Multi_Lick_ports_behavioral_setup). Spike extraction used SpikeSorter (swindale.ecc.ubc.ca/home-page/software). Connectivity estimation used TSPE toolbox (https://github.com/biomemsLAB/TSPE). Estimated firing rate using Bayesian adaptive kernel smoother (https://github.com/nurahmadi/BAKS). Venn diagrams were plot using a customized Matlab script (https://www.mathworks.com/matlabcentral/fileexchange/22282-venn). Other customized codes are available from https://doi.org/10.5281/zenodo.7953792.

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

## Acknowledgements

We thank Dr. Z. Zhang for imaging and C. Zhang for animal husbandry, histology assistance and items purchasing. We thank O. Gonzalez for 3D printing and helps on behavioral set-up design. We thank Drs. K. Gerow, C. Jiang, M. Minear, and J. Dai for discussion. We thank Dr. N. Li for providing Dual Lick Port Detector. AAV-CAG-hChR2-H134R-tdTomato was a gift from Karel Svoboda. pAAV-Ef1a-mCherry-IRES-Cre was a gift from Karl Deisseroth. pAAV-hSyn-DIO-hM4D(Gi)-mCherry was a gift from Bryan Roth. pAAV-CKIIa-stGtACR2-FusionRed was a gift from Ofer Yizhar. This work is supported by grants from National Institute of Mental Health (1R21MH131363-01) and from National Institute of General Medical Sciences (2P20GM121310).

## Author contributions

Q.S. designed and supervised the project, acquired the funding, and edited the manuscript. Y.W. designed the project, built the behavioral set-up, collected and analyzed data, performed the simulation, and wrote the original manuscript.

## Competing interests

The authors declare no competing interests.
