## [Peer Review File · Nature Communications]

A prefrontal motor circuit initiates persistent movementREVIEWER COMMENTS

Reviewer #1 (Remarks to the Author):

This manuscript by Wang et al. investigated neural mechanisms underlying persistence of movement actions. They trained the mouse to perform persistent licking actions (cycling tongue movements for tens of seconds) upon water delivery, which stopped quickly after quinine delivery. Using this task, they could study the initiation and termination phases of persistent licking. They then performed single-unit recording in dmPFC, M1 and insula cortex (IC) and found that neural encoding of movement phases (within a seconds time window) and valence of liquid (within a much shorter time window, associated with each lick) is rather independent. In particular, the motor cortex projecting (MP) neurons in dmPFC mainly encode the initiation phase of movements. Silencing of these neurons impaired the initiation of movements and reduced activity in M1 and IC. The authors also used a computation model of the MP-based network to show that a continuous sensory signal to dmPFC is needed for triggering the persistence. There haven't been many studies on the persistence of movement, so this study provides valuable information for the field. The experimental execution and data analysis are sound. The manuscript, however, has not been well-written. There are numerous grammatical mistakes, and some descriptions are confusing.

Specific points

In the section "Dissociable neural coding of movement phases and valence", are the results described for the entire population including cells in all three regions? "We found that 81.97% of neural representations of initial phase... and 51.41% of neural representations of terminal phase showed weak taste tuning (Fig 1D-E)...". Does this mean that 18% of neural representations of initial phase and 49% of neural representations of terminal phase showed strong taste tuning? These numbers are not directly shown in Fig.1E. Perhaps the authors can plot a separate chart to show the overall fractions of neurons tuned to movement phases versus taste. Suggestion: change "weak" to "weak or no".

"...no overall excitatory impact from initial phase to PV or from PV to initial phase neural representations (mean of real TSPE < 99 percentile of shuffled TSPE, Fig S5E)." Do the authors mean Fig. S5D? There is no Fig. S5E.

In the section "Coding of initial phase in dmPFC MP neurons", it appears that MP neurons were recorded specifically. Here, the authors need to specify in the text how these neurons were identified through optogenetics aided recording (Fig.2A-C). Also, in the previous section, were the dmPFC neurons the same MP population or were they a general dmPFC population without cell-type specificity?

"...the separations of positive and negative value at S- and L- window were compared." What does this mean?

“The results showed that the separation of PCA trajectories at L-window was significant higher than it at S-window”, please add the implication of this result: MP neurons distinguish between movement phases better than between valence values.

“...no significant difference between these two phases ($p > 0.05$, Fig 1M)”, do the authors mean Fig.2L-M?

In Fig.4, how long does it take for M1/IC activity to recover after the termination of opto-inhibition of dmPFC?

In the section of modeling, “56% connectivity is excitatory and 44% of it is inhibitory”, the fraction of inhibitory neurons seems much higher than that in the real cortex. “...we simulated the optogenetic silencing of dmPFC MP neurons through decreasing the number of neurons in the modeled network.” Are the numbers of both excitatory and inhibitory neurons reduced or just excitatory neurons? In the real silencing condition, not only the number of spiking neurons is reduced but also the firing rate of remaining neurons is reduced. Can the authors simulate the silencing condition by adding additional inhibitory currents into affected cells?

The modeling results imply that continuous sensory stimulation is necessary for the persistent output of the MP network. Can the authors speculate what sensory cue serves such a role in this particular task? Since MP neurons poorly encode valence, taste signals are unlikely serving this role.

In Discussion, can the authors also discuss about the gradually increased firing of NV representing neurons (e.g. Fig.S4D, second cell)?

There are numerous mistakes/typos. Just name a few:

Introduction, “Socially dominate animal” changed to “Socially dominant animals”.

First result section, “or the liquid switched to quinine” changed to “or when the liquid switched to quinine”. “We next quantified the examination window...”, here the authors did not quantify but just arbitrarily determined. “...evaluating whether water or quinine is hedonic or aversive should base on the each contact”, changed to “...should be based on...”.

Second section, “To test the possibility that the neural networks of representing movement phases and valence may interact with each other...” changed to “To test whether the neural networks representing movement phases and valence interact with each other...”

“This result is in agreement with previously reported overall dmPFC neurons, which were showed to delay the initiation of avoidance (Jercog et al., 2021).” Change to “..., silencing of which has been shown to delay the initiation of avoidance”.

“To test the hypothesis, we measured the neural activity in these two brain regions by shined with or without laser on dmPFC.” Change to “we measured the neural activity in these two brain regions with or without shining laser on dmPFC.”

Discussion, “Our findings suggest that the decision to proceed to a persistent movement is not relied on an acute judgement of the present situation.” Suggested to change to “...the decision to engage in persistent movement is not based solely on a snap judgment of the current situation.”

“This connection configuration may help to not only simultaneously receive the signals from the neurons that tune multiple aspects of sensory cues...” change to “...tuned to multiple aspects..”

“...it is unclear.... how specific movement execute under a persistent pattern” change to “...how a specific movement is executed...”.

Reviewer #2 (Remarks to the Author):

In this study, Wang and colleagues investigate brain regions involved in persistent motor activity using multi-electrode recordings and viral techniques. They use the rhythmic licking by a head-fixed water-deprived mouse, in a behavioral task to measure the length of the licking as a measure of persistence in motor activity. There are two conditions: negative valence when the water is replaced by a poorly-tasting substance, and a positive valence, where sugar is added to the water. Under these conditions, the authors are interested in the activity in the dorsomedial prefrontal cortex (dmPFC) and the input to the primary motor cortex. To ensure this they inject a virus in M1 that can be carried and expressed in the somata sitting in dmPFC. The virus expresses a photo-sensitive opsin (Channelrhodopsin-2) such that it can identify the extracellular units by Opto-tagging. The authors also investigate the insular cortex, which is believed to represent valences of sensorimotor activity. The work is interesting and thorough, and the model paradigm is clever and suitable for addressing the questions of persistence and valence. The presentation of the work is confusing and disorganized. The manuscript needs a comprehensive rewrite for potential readers to benefit from the work.

Concerns:

My major concern is the lack of systematic presentation and writing of the study. There are many different experiments and approaches, but they are often not presented in an orderly fashion with a clear scope. The line of work needs to be presented clearly in a sequence. This needs to be done to conform to the standard format of scientific papers. For instance, the experiments should be better described in the abstracts, especially the experimental method (multi-electrode recording, awake behavioral/head-fixed and optogenetics, type of animal). Here is a list of some of the improvements that could be done:

I suggest defining what the authors mean by “persistence” at a very early point, preferably in the abstract. Maybe something like the “ability to continue an action”

Please mention exactly what is being measured in the experiment. Is it optical Calcium imaging or extracellularly recorded spikes? This information should also be in the abstract.

Line 25 trait -> property

Line 46: Please briefly explain what quinine is and why it was used.

Line 47 won't -> did not

Line 54 Abbreviation DO is defined later in the text. Should be defined the first time it is used.

Line 104 What is the rationale behind looking at arousal? I don't understand why this experiment is important please summarize the conclusion in one sentence. And why not use the more common and quantifiable measures of arousal (cortical desynchronization)?

Figure 2: PCA: Why not show trajectories of more trials?

Figure 3 A and B are not described in the text anywhere-

Figure 5: Better to write out the abbreviation “DO” in the caption.

Reviewer #3 (Remarks to the Author):

I think the most interesting part of the study is the description of a circuit that integrates sensory input (from the insular cortex) into a central processing brain region (the dmPFC) to modulate motor output (in M1), using a behavioral paradigm involving thirty licks. This study showed that neural activity in dmPFC neurons encodes the initiation of licking but not persistent licking, which is consistent with the decision-making function proposed for the dmPFC in many previous studies. The circuit structure described in the group's previous study shows that the insula projects to the dmPFC, which then projects to M1. Thus, it is confusing that the neural activity of the insula also decreases after silencing dmPFC neurons. It would be helpful to explore the possible mechanism behind this observation. Additionally, in order to complete the circuit, it would be useful to investigate whether the dmPFC-projecting insular neurons serve the same function.

Thirst is an internal state that triggers drinking behavior, which persists until the thirsty state is satisfied. The persistent licking behavior is likely mediated by subcortical neural circuits, and resembles reflex behavior that does not require cortical inputs. Consistent with this, the paper found that M1-projecting dmPFC MP neurons regulate part of the initiation of licking after water deprivation. Therefore, I do not think the "persistence" described in the introduction refers to the same thing as the "persistent" movement or licking described in this study. Additionally, the introduction is too brief to provide sufficient background information on the study. It needs to be revised according to the study's purpose.

For Fig 1D and E, are the recordings from all three regions? I think it would be better to analyze the three regions separately. Also, were the recordings in different brain regions taken from the same or different animals?

Regarding Fig 2, have the authors examined the response of non-MP neurons? Is the correlated activity observed during licking only present in M1-projecting neurons?

I am concerned that a 22% drop in body weight is too severe, and the mice may not recover well.

Both male and female mice were used in the study, but the data collected from males and females were not specified, nor were any differences between the results of males and females reported.

The MP neurons in the dmPFC should be introduced when they are first mentioned.

Regarding Fig 3, it would be helpful to quantify overall licking, as well as licking after the laser.

In the Methods section, it states, "For viral injection, P14-30 mice were used." Why were P14 mice used, and should their brain coordinates be different from those of adults?

The text and legend labels in most supplementary figures are too small to read. For example, in Fig S3 and S4, the results are crucial to evaluate the experiments, but they are difficult to read.

Reviewer #4 (Remarks to the Author):

Wang & Sun performed a series of electrophysiological and behavioral experiments to study what they call “persistence.” Unfortunately, the manuscript is not well written, which makes it difficult to understand their arguments. Just to list a few...

- 1) The definition of “persistence” is unclear. The introduction describes long-timescale persistence for social behaviors etc. In contrast, authors study persistent licking in a relatively short time scale (~10s). Are these relevant? Mechanisms of “persistence” is likely different across timescale.
- 2) Intro: Authors claim “a paucity of studies on its behavioral and neural mechanism”. But if authors refer to persistent spiking activity (or behavior) in a sec-min timescale, it has been extensively studied across species.
- 3) P2, L3: “water was deprived for 16 to 36 hours until body weight decreased around 22%”. It is impossible that body weight decreases this much in 36 hours. Indeed, the method section describes something different...
- 4) The way authors defined the examination window is subjective. For example, why do the authors assume that the movement-related activity appears on a longer time scale than the valance signal? Any evidence for this? Most likely, many neurons show mixed selectivity within the same time window. In addition, do the authors show all brain regions (mPFC, IC, M1) in one plot (Fig1DE)?
- 5) Fig2. I do not understand why the authors do not define and mention how they have recorded “dmPFC MC” in the main text. In addition, the comparison of distance in PC space does not make sense (the number of time bins, the fraction of variance explained, etc., are all different, and the comparison is not meaningful).
- 6) Fig3: why did the authors not try laser on and off within the same animal instead of comparing the laser and sham groups?
- 7) P4 2nd paragraph “We next hypothesized...”: I cannot tell where this hypothesis comes from...There could be many ways to suppress movement initiation, such as suppressing motor command without affecting the valance signal. They should discuss more.
- 8) Model: Why does the model have a persistent input? Then, the persistence comes from the input, not the network dynamics.

We appreciate the Reviewers' feedback and comment, which has helped us identify areas of confusion and improve our manuscript. On the following pages, please find our detailed point-to-point responses. As an overview, we have added the following new analyses to support and interpret our findings, and have substantially revised the manuscript. We hope that reviewers find our responses thorough and that their concerns have been satisfactorily addressed.

1. Comparison of behavioral performance between male and female mice (Supplementary Fig. 10-s).
2. Overall percentage of neural representations in IC, M1, and dmPFC (Fig. 1e).
3. All pca trajectories of dmPFC MP neurons (Fig. 2f, i).
4. Comparison of lick frequency after the laser off (Supplementary Fig. 10c4).
5. Summarized model for how dmPFC MP neurons initiate persistent movement (Fig. 6).

Reviewer #1 (Remarks to the Author):

This manuscript by Wang et al. investigated neural mechanisms underlying persistence of movement actions. They trained the mouse to perform persistent licking actions (cycling tongue movements for tens of seconds) upon water delivery, which stopped quickly after quinine delivery. Using this task, they could study the initiation and termination phases of persistent licking. They then performed single-unit recording in dmPFC, M1 and insula cortex (IC) and found that neural encoding of movement phases (within a seconds time window) and valence of liquid (within a much shorter time window, associated with each lick) is rather independent. In particular, the motor cortex projecting (MP) neurons in dmPFC mainly encode the initiation phase of movements. Silencing of these neurons impaired the initiation of movements and reduced activity in M1 and IC. The authors also used a computation model of the MP-based network to show that a continuous sensory signal to dmPFC is needed for triggering the persistence. There haven't been many studies on the persistence of movement, so this study provides valuable information for the field. The experimental execution and data analysis are sound. The manuscript, however, has not been well-written. There are numerous grammatical mistakes, and some descriptions are confusing.

Specific points

In the section "Dissociable neural coding of movement phases and valence", are the results described for the entire population including cells in all three regions? "We found that 81.97% of neural representations of initial phase... and 51.41% of neural representations of terminal phase showed weak taste tuning (Fig 1D-E)...". Does this mean that 18% of neural representations of initial phase and 49% of neural representations of terminal phase showed strong taste tuning? These numbers are not directly shown in Fig.1E. Perhaps the authors can plot a separate chart to show the overall fractions of neurons tuned to movement phases versus taste. Suggestion: change "weak" to "weak or no".

Our response: The Reviewer's understanding is correct. In the manuscript, "81.97%" and 51.41%" denote the non-overlap percentages of the neural representations of specific movement phases in movement phase tuned neurons, but not all recorded neurons. In the Fig 1e, the numbers in the Venn diagram denote the percentage of specific movement phases in all recorded neurons. To make it clear,

we add an overall plot in the Fig 1e (Reviewer Figure 1.1) and revised the manuscript accordingly. Following is our revision: “We found that 27% of single-units represented movement phases (initial or terminal phase, $\geq 65\%$ in movement phase correlation) but showed weak or no taste tuning (Fig. 1d, e). 14% of single-units represented valence (positive or negative valence (PV or NV), $z > 1.64$ in taste correlation) but exhibited poor specification on movement phases (Fig. 1d, e). By contrast, only small fraction (8%) of them displayed the preference to both valence and movement phases (Fig. 1e), ... ”.

Line# 81-86 Page# 3

As known above, per reviewer suggestion, “weak” has been changed to “weak or no”.

Overall

Reviewer Figure 1.1 (from Fig. 1e) Pie chart shows the percentage of the classified neural representations.

“...no overall excitatory impact from initial phase to PV or from PV to initial phase neural representations (mean of real TSPE < 99 percentile of shuffled TSPE, Fig S5E).” Do the authors mean Fig. S5D? There is no Fig. S5E.

Our response: Thanks for the Reviewer’s correction. We have changed Supplementary Fig. 5e to Supplementary Fig. 5d in the manuscript. Line# 91 Page# 3

In the section “Coding of initial phase in dmPFC MP neurons”, it appears that MP neurons were recorded specifically. Here, the authors need to specify in the text how these neurons were identified through optogenetics aided recording (Fig.2A-C). Also, in the previous section, were the dmPFC neurons the same MP population or were they a general dmPFC population without cell-type specificity?

Our response: We used Stimulus Associated spike Latency Test (SALT) to identify the MP neuron. We insert a citation in the manuscript as following: “Using cell-specific recordings enabled by the opto-tagging approach ¹ (Fig. 2a-c), we found that...”. Line# 100 Page# 3

The dmPFC neurons were the general dmPFC population without cell-type specificity.

“...the separations of positive and negative value at S- and L- window were compared.” What does this mean?

Our response: We are sorry for this confusion. We have replaced this sentence with the revision: “To confirm the discriminability of dmPFC MP neuron between valence and movement phases, we first embedded neural population activity at S- and L-window of dmPFC MP neurons into trajectories by principal component analysis (PCA) and then measured mean Euclidean distances in all PC dimensions.”
Line# Page#

“The results showed that the separation of PCA trajectories at L-window was significant higher than it at S-window”, please add the implication of this result: MP neurons distinguish between movement phases better than between valence values.

Our response: We added this statement to the manuscript. Line# 116-117 Page# 3

“...no significant difference between these two phases ($p > 0.05$, Fig 1M)”, do the authors mean Fig.2L-M?

Our response: Yes, it is Fig.2l, m. we have corrected this part. Line# 129-130 Page# 4

In Fig.4, how long does it take for M1/IC activity to recover after the termination of opto-inhibition of dmPFC?

Our response: To silence the activity of dmPFC MP neurons, we used anion-conducting channelrhodopsins (ACRs), which induce highly efficient silencing when the light is turned on and recover rapidly when the light is turned off. To confirm the recovery, we extended the measurement window to 2 seconds after the laser was turned off. We found that there was no significant difference in the estimated firing rate (Reviewer Figure 1.2, 5s-7s after DO). This indicates that the recovery time is less than 2 seconds. Our interpretation for the separation between the laser and sham curves even after the laser stop in our original figure is that the spike raster data were subject to a Gaussian distribution when estimating the firing rate. When the plot edge is close enough to the laser or sham delivery end, the z-score drop in the edge region of the curves causes them to be separated.

Reviewer Figure 1.2. (from Fig. 4f,g) Neural activity rapidly recovered after the termination of opto-inhibition.

Mean baseline subtracted, z scored firing rate, relative to water DO, of IC (a) and M1 (b) neurons.

In the section of modeling, “56% connectivity is excitatory and 44% of it is inhibitory”, the fraction of inhibitory neurons seems much higher than that in the real cortex. “...we simulated the optogenetic

silencing of dmPFC MP neurons through decreasing the number of neurons in the modeled network.” Are the numbers of both excitatory and inhibitory neurons reduced or just excitatory neurons? In the real silencing condition, not only the number of spiking neurons is reduced but also the firing rate of remaining neurons is reduced. Can the authors simulate the silencing condition by adding additional inhibitory currents into affected cells?

Our response: We fully agree that inhibitory neurons have only a small fraction in the real cortex. However, the detailed connectome is unclear. Normally, the inhibitory neurons have a higher connection probability than excitatory neurons, but we do not know the exact probability. To simulate the neural network, we could either increase the connection probability of inhibitory neurons or increase the number of inhibitory neurons. Overall, the ultimate goal is to create a relatively uniformly distributed spike along the timeline (Reviewer Figure 1.3c). In our case, we set the connection probability of excitatory and inhibitory neurons to 0.1 and 0.2, respectively. Although the proportion of inhibitory neurons may be less than 44%, the connection probability of the inhibitory neuron may be higher than 0.2 and that of the excitatory neuron may be lower than 0.1. If we fix this connection probability, a decrease in the proportion of inhibitory neurons will disturb the uniform distribution pattern of spikes (Reviewer Figure 1.3a-b) and cause it to be different from the real spike distribution in the cortex.

In the scenario of opto-silencing simulation, we decreased the number of overall types of neurons. We have added this illustration in both Figure 5 legend and manuscript.

To simulate additional inhibitory currents, we added an external inhibitory network comprising 100 inhibitory neurons. Triggering the simulated MP network would also trigger this external inhibitory network, resulting in higher inhibition of MP neurons. Our results showed that strong MP inhibition (N = 450) affected the initiation bias (Reviewer Figure 1.3e). However, mild MP inhibition (N = 900) did not affect initiation (Reviewer Figure 1.3e).

Reviewer Figure 1.3 a-c. Raster plots show spike times with different excitatory (E)/inhibitory (I) ratio as indicated. d. Lick raster plot show the simulation of opto-silencing by decreasing neuron number and adding inhibitory currents. e. Comparison of initiation bias generated from various simulated networks.

The modeling results imply that continuous sensory stimulation is necessary for the persistent output of the MP network. Can the authors speculate what sensory cue serves such a role in this particular task? Since MP neurons poorly encode valence, taste signals are unlikely serving this role.

Our response: We suspect this continuous sensory stimulation may come from highly generalized sensory signals in the sensory thalamus because attentional behavior is causally dependent on the interactions of the mPFC with the sensory thalamus². Since persistent movement requires a high degree of concentration, we think that attention is crucial for it. Therefore, the sensory thalamus might play a role in the continuous sensory signaling.

In this particular task, the sensory cue is very likely to be a sensation of liquid delivery. We have added this clarification at the section of modeling.

In Discussion, can the authors also discuss about the gradually increased firing of NV representing neurons (e.g. Fig.S4D, second cell)?

Our response: we have added the discussion at the manuscript as following: "Similarly, the gradually increasing activity of the neural representations of NV (Supplementary Fig. 4d, e) could also be caused by the transformation of their encoding properties or by an increasing negative valuation of water."

Line# 232-234 Page# 6

There are numerous mistakes/typos. Just name a few:

Introduction, "Socially dominate animal" changed to "Socially dominant animals".

First result section, "or the liquid switched to quinine" changed to "or when the liquid switched to quinine". "We next quantified the examination window...", here the authors did not quantify but just arbitrarily determined. "...evaluating whether water or quinine is hedonic or aversive should base on the each contact", changed to "...should be based on...".

Second section, "To test the possibility that the neural networks of representing movement phases and valence may interact with each other..." changed to "To test whether the neural networks representing movement phases and valence interact with each other..."

"This result is in agreement with previously reported overall dmPFC neurons, which were showed to delay the initiation of avoidance (Jercog et al., 2021)." Change to "..., silencing of which has been shown to delay the initiation of avoidance".

"To test the hypothesis, we measured the neural activity in these two brain regions by shined with or without laser on dmPFC." Change to "we measured the neural activity in these two brain regions with or without shining laser on dmPFC."

Discussion, "Our findings suggest that the decision to proceed to a persistent movement is not relied on an acute judgement of the present situation." Suggested to change to "...the decision to engage in persistent movement is not based solely on a snap judgment of the current situation."

"This connection configuration may help to not only simultaneously receive the signals from the neurons

that tune multiple aspects of sensory cues...” change to “...tuned to multiple aspects..”
“...it is unclear.... how specific movement execute under a persistent pattern” change to “...how a specific movement is executed...”.

Our Response: Thanks Reviewer’s feedback. We apologize for these oversight and all these errors have been corrected in the manuscript.

Reviewer #2 (Remarks to the Author):

In this study, Wang and colleagues investigate brain regions involved in persistent motor activity using multi-electrode recordings and viral techniques. They use the rhythmic licking by a head-fixed water-deprived mouse, in a behavioral task to measure the length of the licking as a measure of persistence in motor activity. There are two conditions: negative valence when the water is replaced by a poorly-tasting substance, and a positive valence, where sugar is added to the water. Under these conditions, the authors are interested in the activity in the dorsomedial prefrontal cortex (dmPFC) and the input to the primary motor cortex. To ensure this they inject a virus in M1 that can be carried and expressed in the somata sitting in dmPFC. The virus expresses a photo-sensitive opsin (Channelrhodopsin-2) such that it can identify the extracellular units by Opto-tagging. The authors also investigate the insular cortex, which is believed to represent valences of sensorimotor activity. The work is interesting and thorough, and the model paradigm is clever and suitable for addressing the questions of persistence and valence. The presentation of the work is confusing and disorganized. The manuscript needs a comprehensive rewrite for potential readers to benefit from the work.

Concerns:

My major concern is the lack of systematic presentation and writing of the study. There are many different experiments and approaches, but they are often not presented in an orderly fashion with a clear scope. The line of work needs to be presented clearly in a sequence. This needs to be done to conform to the standard format of scientific papers. For instance, the experiments should be better described in the abstracts, especially the experimental method (multi-electrode recording, awake behavioral/head-fixed and optogenetics, type of animal). Here is a list of some of the improvements that could be done:

Our response: We are grateful to the reviewer’s suggestion. We have reorganized our writing presentations. Here is a list of what we have done.

- (1) We described the experimental approaches in both abstract and results;
- (2) We defined persistence in the first sentence in the abstract; we defined what type of persistent movement we studied in the introduction; we defined the property of persistent movement in the section “Quantification of persistent movement”.

I suggest defining what the authors mean by “persistence” at a very early point, preferably in the abstract. Maybe something like the “ability to continue an action”

Our response: Following the reviewer’s suggestion, we have clarified the persistence in the first sentence of the abstract: “Persistence firms continuance in a course of action.” Line# 10 Page# 1

We added a sentence in the introduction: "To examine persistence, we focus on a motivated movement that can persist for a certain period of time and stabilize at a range of high frequency." Line# 30-31 Page# 1

Please mention exactly what is being measured in the experiment. Is it optical Calcium imaging or extracellularly recorded spikes? This information should also be in the abstract.

Our response: we have added specific experimental approaches orderly in both abstract and results.

Line 25 trait -> property Revised Line# 22 Page# 1

Line 46: *Please briefly explain what quinine is and why it was used.*

Our response: We revised it in the manuscript as following: "As an aversive stimulus, quinine administration was more likely than the interruption of water or sucrose administration to cause termination of persistent licking ($p < 0.05$ with respect to termination bias, Supplementary Fig. 1i). ... Therefore, quinine was used to terminate the persistent licking movement and to evaluate negative valence." Line# 49-53 Page# 2

Line 47 won't -> did not Revised Line# 49 Page# 2

Line 54 Abbreviation DO is defined later in the text. Should be defined the first time it is used. Revised Line# 41 Page# 2

Line 104 *What is the rationale behind looking at arousal? I don't understand why this experiment is important please summarize the conclusion in one sentence. And why not use the more common and quantifiable measures of arousal (cortical desynchronization)?*

Our response: The arousal we concerned about is bodily arousal, which exhibited a short increase of bodily activity after stimuli (Supplementary Fig. 1k-n). Similar with valence, we think bodily arousal could be a potential feature that is coded in a population of neurons, like some neurons in the dmPFC (Supplementary Fig. 8d). To exclude this potential coding in dmPFC MP neuron, we looked at bodily arousal. We added a one-sentence conclusion in the manuscript as following: "These results excluded the possible coding of bodily arousal in dmPFC MP neurons". Line# 130 Page# 4

In contrast, desynchronization of cortical activities is not a behavioral feature and cannot be used to examine the neural coding. This is why we did not consider it. To make this clear in the manuscript, we added a sentence at the beginning of this paragraph: "Bodily arousal is usually accompanied by valence changes. It is, therefore, necessary to investigate whether or not dmPFC MP neuron also encode bodily arousal." Line# 118-119 Page# 3

Figure 2: PCA: Why not show trajectories of more trials?

Our response: Since we plotted two trajectories simultaneously, we were concerned that showing all trials might make the plot less clear. However, as requested, we have replaced the original figure with the plot that shows more trials (Reviewer Figure 2.1).

Reviewer Figure 2.1 (from Fig. 2f,i) a. PCA trajectories of dmPFC MP neurons. b. bar plot showing cumulative variance explained percentage of first two PCs. Values are mean \pm s.e.m.

Figure 3 A and B are not described in the text anywhere-

Our response: We have added the description as following “dmPFC MP neurons were optogenetically manipulated by expressing stGtACR2 and shining laser during the different phases of the persistent licking task (Fig. 3a). Our results showed that optogenetic silencing of dmPFC MP neurons impaired the initiation of licking ($p < 0.001$ compared to the sham trials, Fig. 3b, c)...” at the section “Effect of dmPFC MP neuron silencing on persistent movement”. Line# 134-137 Page# 4

Figure 5: Better to write out the abbreviation “DO” in the caption.

Our response: We have corrected this part. Line# 564 Page# 15

Reviewer #3 (Remarks to the Author):

I think the most interesting part of the study is the description of a circuit that integrates sensory input (from the insular cortex) into a central processing brain region (the dmPFC) to modulate motor output (in M1), using a behavioral paradigm involving thirty licks. This study showed that neural activity in dmPFC neurons encodes the initiation of licking but not persistent licking, which is consistent with the decision-making function proposed for the dmPFC in many previous studies. The circuit structure described in the group's previous study shows that the insula projects to the dmPFC, which then projects to M1. Thus, it is confusing that the neural activity of the insula also decreases after silencing dmPFC neurons. It would be helpful to explore the possible mechanism behind this observation. Additionally, in order to complete the circuit, it would be useful to investigate whether the dmPFC-projecting insular neurons serve the same function.

Our response: Based on others³ and our study (Fig. 2k), insular cortex (IC) encode taste valence. Although IC projects to motor cortex, there is little or no innervation from the positive valence (PV) to initial phase tuned neurons (Supplementary Fig. 5d). Therefore, persistent licking should not be

triggered by the signal of the PV. On the contrary, the PV signal should be induced only after the animal contacting with the liquid. For this reason, IC and M1 neuronal activity decreased after the dmPFC MP neuron was inactivated. As for the reason why IC projects to dmPFC MP neuron, we think it is probably because the MP projecting neurons in the IC do not encode taste valence but initial persistent movement phase (Supplementary Fig. 4a2).

To discuss it, we added a paragraph in the Discussion as following: **“The function role of MP neuron in IC-MP-motor circuit.** Based on the circuit structure, MP neuron acts as a node connecting IC and motor region. Since IC encodes valence, it is possible that the MP neuron integrates valence signal and sends it to motor region. However, our result showed that MP neuron does not encode valence (Fig. 2k). Because some of the neurons in IC also encode movement phases (Fig. 2k and Supplementary Fig. 4a), we speculate the function of this circuit is to transmit movement phase signals, instead of valence, during persistent movement control.” Line# 235-240 Page# 6

To further clarify this point, we have added a summarized plot (Reviewer Figure 3.1).

Finally, we agree with the reviewer that it would be useful to investigate whether the dmPFC-projecting insular neurons serve the same function. However, to answer this questions, a new subset of studies that are similar with how we study MP neurons in the dmPFC during different phases of the behavior task is required for the mPFC projecting IC neurons. Such studies are beyond the scope of the current hypothesis.

Reviewer Figure 3.1 (from Fig 6) Summarized model for how dmPFC MP neurons initiate persistent movement.

a. Pattern of neuronal activation during the persistent licking movement. In the initial phase (lick frequency is driven to the maximum), the indicated three types of neurons are activated. In the persistent phase, dmPFC MP neurons are inactivated, while M1 and IC neurons show persistent activity.

b. During the initial phase, when the sensory signals are received, dmPFC MP neurons activate themselves and the downstream motor neurons. The latter drive the licking movements that bring the tongue into contact with the water. The taste signal from the tongue is further transmitted to the

valence-encoding neurons in the IC. During the persistent phase, no signal is transmitted from the dmPFC MP neurons to the motor neurons, but the motor neurons is continuously activated. This results in persistent licking movement and persistent valence encoding in the IC.

Thirst is an internal state that triggers drinking behavior, which persists until the thirsty state is satisfied. The persistent licking behavior is likely mediated by subcortical neural circuits, and resembles reflex behavior that does not require cortical inputs. Consistent with this, the paper found that M1-projecting dmPFC MP neurons regulate part of the initiation of licking after water deprivation. Therefore, I do not think the "persistence" described in the introduction refers to the same thing as the "persistent" movement or licking described in this study. Additionally, the introduction is too brief to provide sufficient background information on the study. It needs to be revised according to the study's purpose.

Our response: First, we agree with the reviewer that persistent licking is mediated by subcortical neural circuits. After all, persistent licking cannot occur if the subject is not thirsty. However, the experimental result that the licking behavior involves increases in neuronal activity in almost all brain regions, including cortical regions ⁴ argues against the idea that thirst-mediated licking is a simple reflex behavior. Second, persistent lick is different from occasional lick. Neural encoding of licking occurs mainly in the motor cortex (12% neural representation of licking, Supplementary Fig. 4b1), but persistent licking requires dmPFC MP neuron. Third, persistent licking requires 2-3 times of training. Thirsty, naive mice did not perform a persistent licking (Reviewer Figure 3.2), even if they were aware that water was delivered from the lick ports (see Reviewer Video 3.1 & 3.2 for the mouse with not well trained and well trained). Taken together, these evidence indicate that the persistent lick is an active, motivated behavior that requires the coordination of multiple brain functions.

Reviewer Figure 3.2 Lick frequency evolving with time between well trained and not well trained thirsty mouse.

To clarify what is being studied, we add a paragraph in the introduction as following: "...the psychological definition of persistence is not appropriate to solve certain behavioral problems, including how long it should last and how to quantify its continuity, as it can vary from behavior to behavior. To examine persistence, we focus here on a motivated movement ⁴ that can persist for a certain period of time and stabilize at a range of high frequency. This type of motivated movement is triggered by internal states from deep brain regions and is regulated by cortex neurons. It is also accompanied with extensive valence coding in the brain. Here, we attempt to identify the neural coding in the different phases of persistent movement and to test whether it is the same with the coding of valence." Line# 28-35 Page#

For Fig 1D and E, are the recordings from all three regions? I think it would be better to analyze the three regions separately. Also, were the recordings in different brain regions taken from the same or different animals?

Our response: Yes, the recordings were from all three regions. We have also analyzed these three regions separately (Supplementary Fig. 4, Supplementary Fig. 6, and Supplementary Fig. 7). We have three types of recording electrode implants: dmPFC only; IC and M1; IC only. Neural activity data are collected from these three types electrodes implant mice.

Regarding Fig 2, have the authors examined the response of non-MP neurons? Is the correlated activity observed during licking only present in M1-projecting neurons?

Our response: We have examined the overall response of dmPFC neurons. The correlated activity data is presented in Supplementary Fig. 4c, Supplementary Fig. 6c, and Supplementary Fig. 7c. Briefly, the overall dmPFC neurons present high initial movement phase coding compared with IC and M1. However, we also found that some dmPFC neurons have different properties with MP neurons. 1. Facial activity coding (Supplementary Fig. 8d); 2. Some valence coding (Supplementary Fig. 4c1);

I am concerned that a 22% drop in body weight is too severe, and the mice may not recover well.

Our response: Based on our observation, 22% drop in body weight did not result in unhealthy condition in mice. Generally, mice were fully recovered to their body weight in one or two days after the water was re-supplied. To minimize unwanted effects on health, the subsequent water deprivation was only conducted at least five days since the last experiment.

Both male and female mice were used in the study, but the data collected from males and females were not specified, nor were any differences between the results of males and females reported.

Our response: In response to reviewer's and editorial request, we have compared the sex difference on behavioral test. Our results showed there is no significant difference between male and female mice (Reviewer Figure 3.3).

Reviewer Figure 3.3 (from Supplementary Fig.1r-s) Comparison of initiation bias (r) and termination bias (s) between male and female mice in water-quinine session. n.s. not significant.

The MP neurons in the dmPFC should be introduced when they are first mentioned.

Our response: We made added another introduction in the section “coding of initial phase in dmPFC MP neurons”, in addition to what has already been introduced in the abstract regarding MP neurons.

Regarding Fig 3, it would be helpful to quantify overall licking, as well as licking after the laser.

Our response: In Fig 3, the overall time window is total 15 seconds after water delivery onset. We plot 10s because it is clear to reflect the effect of shining laser. However, we know there might be an effect to animal’s behavior after the laser termination. To show this data, we analyzed the lick frequency at total 15 seconds period. Our data showed that the lick frequency rapidly returned to the same level as sham trials (Reviewer Figure 3.4) after the laser termination.

Reviewer Figure 3.4 (from Supplementary Fig.10c4) lick frequency relative to DO. The analyzing window was extended to full time period. No significant difference between laser and sham treatment was found after laser off. Two sample t-test: n.s. not significant **p<0.01

In the Methods section, it states, "For viral injection, P14-30 mice were used." Why were P14 mice used, and should their brain coordinates be different from those of adults?

Our response: Yes, it is different from adult mice. However, this difference is vary from region to region. For lateral deep brain regions, such as the insular cortex and amygdala, the difference is large. For some middle brain regions, such as the medial prefrontal cortex and primary motor cortex, there is little difference. Since we only injected into primary motor cortex in this study, the brain coordinate is not an issue for us.

The text and legend labels in most supplementary figures are too small to read. For example, in Fig S3 and S4, the results are crucial to evaluate the experiments, but they are difficult to read.

Our response: We apologize for this oversight. Images with higher resolution and larger font size is uploaded.

Reviewer #4 (Remarks to the Author):

Wang & Sun performed a series of electrophysiological and behavioral experiments to study what they call “persistence.” Unfortunately, the manuscript is not well written, which makes it difficult to understand their arguments. Just to list a few...

1) *The definition of “persistence” is unclear. The introduction describes long-timescale persistence for social behaviors etc. In contrast, authors study persistent licking in a relatively short time scale (~10s). Are these relevant? Mechanisms of “persistence” is likely different across timescale.*

Our response: In the tube rank test for social dominance⁵, the time period only last for 10 to 20 seconds. In our experiment of water-quinine session, the persistent licking last for about 15 seconds. Additionally, an increase and decrease of synaptic efficacy in the dmPFC caused an upward and downward movement in the social rank, respectively⁶. Both the time period and the test brain region are relevant to our experiment. However, we agree with the reviewer that the duration of persistence may be vary from behavior to behavior. For example, persistent running can last for an hour according to our observation. Nevertheless, this variance does not affect our conclusion that dmPFC MP neuron is responsible for initiating persistent movement. This is because regardless of how long the persistent behavior may last, the triggering mechanism may still be the same.

We have clarified this part in the introduction as following: “...the psychological definition of persistence is not appropriate to solve certain behavioral problems, including how long it should last and how to quantify its continuity, as it can vary from behavior to behavior. To examine persistence, we focus on a motivated movement that can persist for a certain period of time and stabilize at a range of high frequency. This type of motivated movement is triggered by internal states from deep brain regions and is regulated by cortex neurons. It is also accompanied with extensive valence coding in the brain. Here, we attempt to identify the neural coding in the different phases of motivated, persistent movement and to test whether it is the same with the coding of valence.” Line# 28-35 Page# 1

2) *Intro: Authors claim “a paucity of studies on its behavioral and neural mechanism”. But if authors refer to persistent spiking activity (or behavior) in a sec-min timescale, it has been extensively studied across species.*

Our response: As we stated in abstract, we actually study persistent movements, but not persistent spiking activity. We are sorry for the confusion. We have changed the sentence as “Although persistence is influential in on both humans and animals, there are few studies on how the brain applies it to movements.” Line# 27-28 Page# 1

To clarify this point, we also changed the title as “A prefrontal motor circuit initiates persistent movement”. Different from the persistent spiking activity in prefrontal cortex during the working memory task⁷, we did not observe the persistent spiking activity in dmPFC MP neurons. Instead, the increased neural activity were only in the initial phase (Fig. 2g).

3) *P2, L3: “water was deprived for 16 to 36 hours until body weight decreased around 22%”. It is impossible that body weight decreases this much in 36 hours. Indeed, the method section describes something different...*

Our response: The deprivation time varied from mouse to mouse. For some mice, overnight deprivation can reach 25% body weight decrease, while for others, it requires approximately 2 days to reach this number. We are certain for this time period is correct under our experimental condition. In the methods, we also pointed out the same detail in the section “Subject details”. Line# 263-264 Page# 7

4) *The way authors defined the examination window is subjective. For example, why do the authors assume that the movement-related activity appears on a longer time scale than the valance signal? Any*

evidence for this? Most likely, many neurons show mixed selectivity within the same time window. In addition, do the authors show all brain regions (mPFC, IC, M1) in one plot (Fig1DE)?

Our response: For small scale window, we designed it based on the evidence of tongue movement cycle (Reviewer Figure 4.1a). Since lick frequency was around 6-7 Hz, one cycle of tongue movement is around 144-166 ms. To include all possible coding (lick and valence), 180ms around the lick onset (the touch point during the lick cycle) was chosen for the small scale window. For large scale window, we designed it based on the period of peak lick frequency (Reviewer Figure 4.1b). We speculated that there should be an additional neural coding to increase the lick frequency at this period. Since this period last about 3 seconds (Reviewer Figure 4.1b), a total 5 seconds around the lick onset (plus 2 seconds baseline before the lick onset) was chosen. We are sorry for its confusion. We have clarified it in the manuscript. Line# 59-69 Page# 2

Reviewer Figure 4.1 a. (from Supplementary Fig. 11b) Representative image shows mouse tongue during the lick task. The arrow represents the direction of tongue movement. b. (from Supplementary Fig. 1c) Lick behavior relative to the 1st lick onset in the water(15s)-quinine(15s) session. Inset plot shows that magnified peak lick frequency.

We did find mixed selectivity within the same small scale window (similar decoding accuracy in water-sucrose and water-quinine session in dmPFC (Supplementary Fig. 6c2, c3)). However, it does not affect the evaluation of the coding property of the dmPFC MP neurons. Moreover, the observation that only small fraction of mix selectivity (Fig 1E, 8% overlap) suggests this method is acceptable.

In Fig. 1d, e, the data is from all recorded brain regions.

5) Fig2. I do not understand why the authors do not define and mention how they have recorded “dmPFC MC” in the main text. In addition, the comparison of distance in PC space does not make sense (the number of time bins, the fraction of variance explained, etc., are all different, and the comparison is not meaningful).

Our response: We used a previously defined method to record dmPFC MP neuron. We have added a sentence with a reference as following: “Using cell-specific recordings enabled by opto-tagging approach¹...” Line# 100 Page# 3

It is true that the number of time bins and the fraction of variance explained are different between small- and large-scale windows. However, we did not compare the absolute distance but mean Euclidean distance (as we indicated in the figure legend). Under this condition, the difference of time

bins is not relevant. Despite the fractions of variance explained are different in the first two dimensions, Euclidean distances were calculated not based on only first two dimensions but all dimensions. Under this condition, the fractions of variance in both small- and large-scale windows equal 1. To make it clear, we changed the figure legend as following: "Comparison of mean Euclidean distance from all PC dimensions of dmPFC MP neurons..."

6) *Fig3: why did the authors not try laser on and off within the same animal instead of comparing the laser and sham groups?*

Our response: What we mean the laser and sham group is different trial groups but not different mice. Therefore, it is equivalent to laser on and off. We are sorry it is confusing, we have replaced word "sham group" with "sham trial".

7) *P4 2nd paragraph "We next hypothesized...": I cannot tell where this hypothesis comes from...There could be many ways to suppress movement initiation, such as suppressing motor command without affecting the valance signal. They should discuss more.*

Our response: We have revised the sentence as following: "We next hypothesized that the increase of the brain state of positive valence and tongue movement at initial phase is due to coding of movement phases." Line# 159-160 Page# 4

Besides movement phases, we summarized four features that can possibly suppress the initiation of persistent movement. 1. Valence; 2. Bodily arousal; 3. Motor execution; 4. Motor planning. For the first two features, we have excluded their possible coding in dmPFC MP neuron (Fig. 2). For the motor execution, since suppressing dmPFC MP neuron did not impair bodily activity and lick frequency during the middle phase (Supplementary Fig.10b), the possibility of coding of motor execution in dmPFC MP neuron can be excluded. For the motor planning, if dmPFC MP neuron encode this feature, then increase of neural activity should emerge before the first lick onset. However, most of them emerge after the first lick onset (Fig. 2g). These details can be found in the manuscript. To summarize it, we have also added a sentence in the first paragraph of Discussion section as following: "Our results excluded the possible neural coding, including valence, bodily arousal, and motor command, in dmPFC MP neurons." Line# 198-200 Page# 5

8) *Model: Why does the model have a persistent input? Then, the persistence comes from the input, not the network dynamics.*

Our response: We are sorry it is confusing. This is about the duration of persistence. For the continuous input, we think it should endure about 5 seconds, so does the MP network. Then, the persistent network dynamics can be maintained in motor cortex (Supplementary Fig. 3e, left) for 15-30 seconds. During this time, MP network is not required. In summary, MP network is responsible to start but not maintain persistent movement. In other words, MP network transform the short-term external input into a long-term internal persistence.

In response to question 1), 2), and 8), we added a paragraph in the Discussion section as following: "**Relationships to persistent spiking activity.** The persistent spiking activity in prefrontal cortex has been observed in both rodent and human during the maintenance of working memory^{7,8}. However, few neurons showed persistent activity in prefrontal cortex (small fraction of neural representations of lick, Supplementary Fig. 4c, and gradually decreasing firing of neural representations of PV and initial phase,

Supplementary Fig. 4d, e) during persistent licking. Rather than maintain the neural activity, dmPFC MP neurons are responsible for triggering such activity in the motor cortex. Because this triggering process also requires a successive, short-term (about 5s) external stimuli, we summarize the function of dmPFC MP network as converting a relatively short-term, external, persistent input into a relatively long-term, internal, persistent activity.” Line# 241-249 Page# 6-7

Reference

1. Kvitsiani D, Ranade S, Hangya B, Taniguchi H, Huang JZ, Kepecs A. Distinct behavioural and network correlates of two interneuron types in prefrontal cortex. *Nature* **498**, 363-366 (2013).
2. Wimmer RD, Schmitt LI, Davidson TJ, Nakajima M, Deisseroth K, Halassa MM. Thalamic control of sensory selection in divided attention. *Nature* **526**, 705-709 (2015).
3. Wang L, *et al.* The coding of valence and identity in the mammalian taste system. *Nature* **558**, 127-131 (2018).
4. Allen WE, *et al.* Thirst regulates motivated behavior through modulation of brainwide neural population dynamics. *Science* **364**, 253 (2019).
5. Zhou T, *et al.* History of winning remodels thalamo-PFC circuit to reinforce social dominance. *Science* **357**, 162-168 (2017).
6. Wang F, Zhu J, Zhu H, Zhang Q, Lin Z, Hu H. Bidirectional control of social hierarchy by synaptic efficacy in medial prefrontal cortex. *Science* **334**, 693-697 (2011).
7. Bolkan SS, *et al.* Thalamic projections sustain prefrontal activity during working memory maintenance. *Nat Neurosci* **20**, 987-996 (2017).
8. Kaminski J, Sullivan S, Chung JM, Ross IB, Mamelak AN, Rutishauser U. Persistently active neurons in human medial frontal and medial temporal lobe support working memory. *Nat Neurosci* **20**, 590-601 (2017).

Reviewers' comments:

Reviewer #1 (Remarks to the Author):

The authors have successfully addressed my concerns. I recommend publication.

Reviewer #3 (Remarks to the Author):

The authors have addressed my comments.

Reviewer #4 (Remarks to the Author):

I appreciate the revision and the author's responses, but the manuscript still needs improvement.

Major comments

1) L12: "Here we demonstrate that persistence arises during the initial phase of movement and is maintained until terminal signaling." Is not this the definition of persistent movement (instead of something authors have "demonstrated")?

2) L27 "Although persistence is influential in both humans and animals, there are few studies on how the brain applies it to movements." I appreciate that the authors describe more about "persistence". Yet, this has been extensively studied from the molecular to network levels (see this classic review: <https://pubmed.ncbi.nlm.nih.gov/15582368/>). Most persistent activity literature is in the context of movement, including the classic oculomotor integrator and the persistent escaping behavior of lamprey (i.e., mechanisms of persistent movement have been well-established). Authors should appreciate classic research and significantly modify the introduction to clarify what is new in their research.

3) Water restriction protocol. I am still confused that mice lost 22% of their body weight in 16-36 hours as it significantly deviates from literature often cited for water restriction protocols (<https://pubmed.ncbi.nlm.nih.gov/23849404/> and <https://pubmed.ncbi.nlm.nih.gov/24520413/> ; note that weight only decreased ~10% on day1 of water restriction in these papers; in addition, it claims “however, deprivation in excess of 24 h is not recommended in light of apparent animal distress”). So, even if the number authors claim is correct, animals must be under severe and unnecessary distress due to sudden weight loss, which is concerning for scientific interpretability and also from animal welfare perspectives. 36 hours of water deprivation is extreme. Since there are well-established milder water restriction protocols, authors must explain why they had to select this extreme protocol (GUIDE 8th edition, P30-31 claims “The objective ...should be to use the least restriction necessary to achieve the scientific objective while maintaining animal well-being”).

4) L61: “so we suspected that there was additional neural coding to increase and decrease licking frequency besides the coding of licking command”. It is unclear what this means. What is the definition of “command” here?

5) In Fig1d bottom (and also in Fig2d), many of what authors claim as “valence” signal changed before the lick onset. How is this possible if this is a response to water/quinine?

6) L94: Is it surprising that movement and valence are encoded separately? If so, please explain. If not, what is the point of this analysis?

7) L100: MP-dmPFC appears out of the blue. The comparison of subtype (MP-dmPFC neurons) vs. anonymous cell types in 3 brain areas is an apple-to-orange comparison. The authors should discuss why they have done this cell-type-specific recording.

8) L114: “Euclidian distance in all PC dimensions”: this should be identical to Euclidian distance w.o. PCA...And if this is the main conclusion, why is PCA even necessary?

9) Fig3: Did you have control animals w.o. GtACR expression but with laser? How do you exclude the possibility that mice did not lick because they were startled by blue light? In addition, the power of blue light is not described in Methods.

10) Fig4: It is possible that IC/M1 activity changed because animals did not lick due to MP-dmPFC manipulation. Then, this could be an indirect effect. Can authors analyze trials with and without a lick separately (and also align activity to the first lick)? In addition, this requires control of mice without GtACR expression (see comment 9).

11) L186: Why does decreasing the number of neurons model the silencing of MP-dmPFC neurons? Isn't it more appropriate to inject a negative current into all neurons?

12) L193: Behavior in the persistent lick task is persistent in the sense that it lasts for many seconds, but the mechanism of persistence is the duration of sensory input, according to this sentence (i.e., nothing to do with mechanisms in the brain). Is this the take-home message? If not, the authors should rephrase it.

13) L199: What is the logic for excluding "motor command"?

Minor comment

14) Line 174: What is an "MP network"? Does it mean a network of MP-dmPFC neurons?

We appreciate the Reviewers' feedback and comment. We thoroughly revised the abstract and introduction. Figures were revised as follows: Fig. 2I and Fig. 3f-i. Please find our research rationale and detailed point-to-point responses as follows:

Research rationale

To clarify our research, we made two hypotheses (H1 & H2) on the persistent licking task. H1: the mPFC utilizes the valence signal to drive movement. H2: the mPFC utilizes contextual information to direct movement. In the H1, the mPFC receives the valence signal from each time the tongue touches the liquid and uses this valence signal to trigger another round tongue movement (**Reviewer Figure 1**). In the H2, the mPFC receives continuous contextual information (probably from the sensation of liquid delivery or the sensation of the movement from itself) and uses this signal to initiate movement (**Reviewer Figure 1**).

Reviewer Figure 1. Two hypotheses on the persistent licking task Created with BioRender.com released under a Creative Commons Attribution-NonCommercial-NoDerivs 4.0 International license <https://creativecommons.org/licenses/by-nc-nd/4.0/deed.en>.

Our results support H2. Here is the reason: valence signal is discrete (the liquid is not always on the tongue), and sensory signal is continuous. According to our simulation, discrete input signal will significantly decrease the lick frequency. Therefore, in our study, we provide a specific circuit, which is dmPFC -> motor cortex/striatum, driven by continuous sensory inputs to mediate persistent drinking. If there is opposing valence (e.g. quinine), other circuit(s) will mediate the behavioral switch.

Point-by-point response to the reviewer #4

1) L12: “Here we demonstrate that persistence arises during the initial phase of movement and is maintained until terminal signaling.” Is not this the definition of persistent movement (instead of something authors have “demonstrated”)?

Our response: This is not the definition. For the definition of persistent movement in this study, please see the first sentence in the **behavioral quantification** as following: “First, we defined a persistent movement as the continuous repetition (>6 Hz) of a single movement (e.g., a cycle of tongue or limb movements) and the maintenance of that continuity for at least 5 seconds.”

2) L27 “Although persistence is influential in both humans and animals, there are few studies on how the brain applies it to movements.”. I appreciate that the authors describe more about “persistence”. Yet, this has been extensively studied from the molecular to network levels (see this classic review: <https://pubmed.ncbi.nlm.nih.gov/15582368/>). Most persistent activity literature is in the context of movement, including the classic oculomotor integrator and the persistent escaping behavior of lamprey (i.e., mechanisms of persistent movement have been well-established). Authors should appreciate classic research and significantly modify the introduction to clarify what is new in their research.

Our response: We appreciate the classical literature reviewer brought to our attention. This literature reviewed the generation mechanism of persistent neuronal activity on cellular and network level. However, we did not study the generation of persistent neural activity by internal circuit. Instead, we studied how the animal is motivated to start a persistent movement by selecting specific information, as we described in the title. Such motivation is driven by external input instead of internal dynamics of cell and circuit. Although the neural mechanism on how the action is driven by external input, such as motor planning and execution, has been investigated, the relevant part on persistent movement is currently blank. In general, our study bridges the gap between the study of motivation or decision-making and persistent movement.

As we mentioned in the article, the dmPFC MP neurons are only functional in the initial phase of persistent movement (**Fig. 3**). In the rest of phases, it is possible that the persistent movement is driven by internal persistent neural activity in the primary motor cortex and spinal cord^{1,2}. Nevertheless, such persistent neural activity is different from the persistent neuronal activity at the single cell level which is found in the maintenance of working memory³ (ref Fig. 2 delay phase). In general, the neural activity that maintains persistent movement can be understood as a continuous attractor or rotational dynamics in the network level. Whereas persistent neuronal activity is a continuously firing action potential.

A previous work indicated that the speed of a periodical movement is controlled by speed and brake cells⁴. Therefore, it may require upstream neurons to excite these speed cells so that the movement can reach a high-speed level and stabilize at this level, and we called such movement as persistent movement. Since the persistent movement maintains its speed at a high level, the relevant motor neurons are required to fire periodically in a high density. As such, their activity shows a persistent

pattern, which may be observed in the context of persistent escaping behavior of lamprey. Our study investigated their upstream triggering mechanism.

3) Water restriction protocol. I am still confused that mice lost 22% of their body weight in 16-36 hours as it significantly deviates from literature often cited for water restriction protocols (<https://pubmed.ncbi.nlm.nih.gov/23849404/> and <https://pubmed.ncbi.nlm.nih.gov/24520413/> ; note that weight only decreased ~10% on day1 of water restriction in these papers; in addition, it claims “however, deprivation in excess of 24 h is not recommended in light of apparent animal distress”). So, even if the number authors claim is correct, animals must be under severe and unnecessary distress due to sudden weight loss, which is concerning for scientific interpretability and also from animal welfare perspectives. 36 hours of water deprivation is extreme. Since there are well-established milder water restriction protocols, authors must explain why they had to select this extreme protocol (GUIDE 8th edition, P30-31 claims “The objective ...should be to use the least restriction necessary to achieve the scientific objective while maintaining animal well-being”).

Our response: Our protocol is based on the second indicated literature, which keeps mice weight at 80%. The difference is that we did not keep this number for a long time. Instead, we let the mice recover their weight until the next experiment. In short, our protocol is not a strictly acute water deprivation but a repeated water deprivation. We used this protocol because we noticed it has lightly higher locomotor activity than the chronic one. This is more appropriate to generate stable persistent behavior. To evaluate this protocol, we provide the health data as follows:

Mice health data

According to our observations, the body weight of the mice increased in the long term after we started this protocol (average $8.31\% \pm 1.9\%$ at 60th day, **Method figure 1a**). Similar weight gain was also observed in normal and mild water restricted laboratory rodent⁵. Mice did not perform significantly decreased locomotor activity after water deprivation (**Method figure 1d**), which suggests that mice were not in distress. This result is different from one-time acute water deprivation, which is caused apparent distress when excess 24 hours⁶. We reasoned that mice can adapt to regular acute water deprivation.

Compare with other protocols

Although water deprivation longer than 24 hours is not recommended⁶, this time limit largely depends on the individual conditions⁷. Indeed, water deprivation time is significantly various from mouse to mouse (**Method figure 1c**) to acquire approximately 22% weight loss. Besides body weight, the water deprivation time may also depend on the body water percentage, calorie consumption, nocturnal/diurnal deprivation time ratio, et al, because there is no significant linear correlation between body weight and water deprivation time (**Method figure 1b**). Therefore, the 24-hour time limit is not fixed. As for the percentage of body weight loss, weight loss greater than 15% is also not recommended⁶. However, it may also depend on whether the mice are deprived of water once or several times. Based on a widely used deprivation protocol, mice can remain healthy for four months even after their body weight has stabilized at about 80% of body weight⁸. It suggests that mice can adapt to a new stressful environment.

Based on the body weight data, our protocol is even milder than the indicated second literature. It is more appropriate in our research.

Method figure 1. a. Mice body weights before water deprivation across experimental times. The lines with different colors represent different mouse. **b.** Correlation of the body weights before water deprivation and the 22% body weight loss required water deprivation time. **c.** Comparison of 22% body weight loss required water deprivation time across mice. One-way ANOVA, $F=30.453$, $df=9$, $p=3.5688e-19$. **d.** Comparison of average locomotor activity (represented by average speed) between the mice with and without water deprivation. Two-sample t test, $t=-15936$, $df=22$, $p=0.1253$.

These data were also attached to the **Methods**.

As for why it is different from the previous report that weight only decreased $\sim 10\%$ on day1 of water restriction, we think it is probably because (1) they use CD-1 mouse while we used C57 mouse; (2) their sample size is larger than us (192 vs 10). Because of the variety of body weight loss, it is possible that our data deviates from the averaged data; (3) their experiment is conducted at the sea level in Florida while our experiment is conducted at a higher altitude ($\sim 7700\text{ft}$). It is well documented in human and animal literature that the same amount of water deprivation is likely to result in more body weight loss at high altitudes compared to lower altitudes^{9, 10, 11, 12}. This is due to several factors associated with high-altitude environments: increased respiratory water loss, higher urine production, higher basal metabolic rate, etc. This explains why the same, or even shorter acute water deprivation caused larger weight loss (22% vs. 10-15%). Despite a relatively larger percentage of body weight loss, our animals adapted to the high-altitude environment and acute water deprivations and remained healthy (see above health data).

4) L61: “so we suspected that there was additional neural coding to increase and decrease licking frequency besides the coding of licking command”. It is unclear what this means. What is the definition of “command” here?

Our response: There are types of motor signals: one is from premotor or frontal cortex, which encode latent motor variables; another one is from primary motor cortex, which encode movement kinematics at the individual level¹³. The ‘command’ we mentioned here refers to the motor signal at the level of

individual licks. Because of the frequency difference, other latent motor variables may also emerge in the initial phase.

5) In Fig1d bottom (and also in Fig2d), many of what authors claim as “valence” signal changed before the lick onset. How is this possible if this is a response to water/quinine?

Our response: Lick onset refers to the time point when mouse tongue touches lick port. We set the short time window as lick onset-100ms to lick onset+80ms because we believe the mouse tongue has already touched the water before touching the lick port (**Reviewer Figure 1 H1**). This is why valence signal changes before the lick onset.

6) L94: Is it surprising that movement and valence are encoded separately? If so, please explain. If not, what is the point of this analysis?

Our response: Our results do not imply mean that movements do not correlate with valence under any circumstance. For example, when the thirst level is not high enough, the animal prefers not to lick persistently. Under this circumstance, licking frequency is correlated with valence and MP neurons may also encode valence. However, when the mice are thirsty enough, the valence will be temporally ignored because it cannot drive the licking frequency to the maximized level, as suggested by our computational model. Please see also our hypotheses (**Reviewer Figure 1**). If movement and valence are encoded mixed, then we should accept H1. However, we believe that H2 stated above explains our results better. If mice need to evaluate the liquid type every time they touch the water, then the lick frequency will be much lower than 8 Hz. Of course, if there is an opposite valence, such as quinine, the mice will re-schedule their behavior (i.e. to stop licking). This re-scheduling may take half to one second, during which mice continued to lick the quinine for couple times after the water but did not stop immediately. This supports H2 that movement and valence are encoded separately in a persistent movement.

7) L100: MP-dmPFC appears out of the blue. The comparison of subtype (MP-dmPFC neurons) vs. anonymous cell types in 3 brain areas is an apple-to-orange comparison. The authors should discuss why they have done this cell-type-specific recording.

Our response: Thanks for this question. As we revised in the introduction, since MP-dmPFC neurons innervate motor cortex directly, it is mostly possible that they affect persistent movement. Another reason is that MP-dmPFC connected IC (encodes valence) and M1 (directs movement). This makes MP-dmPFC as a best candidate to test H1. This description is given in Line 32-35.

As for the reason why we have done the cell-type-specific recording in dmPFC while done non-specific recording in dmPFC, IC, and M1 (Supplementary Fig. 4, 6, 7) is because we wanted use the encoding property in other three regions to compare it with MP-dmPFC so that we could know how the information flows from IC to dmPFC to M1 (the unidirectional structure was demonstrated in our previous work¹⁴). Specifically, IC encodes valence based on previous works¹⁵ and our data (supplementary Fig. 4a). We was wondering whether MP-dmPFC also encode valence as IC. The results were negative (Fig. 2h-j). The recordings in dmPFC revealed that the valence faded along the time evolving (supplementary Fig. 6c). It suggests a valence filtering mechanism in dmPFC or in IC. The recordings in M1 is to test whether the signal of valence is filtered out so that the valence will not be used for instructing movement. The results were as expected (Supplementary Fig. 6b). Collectively, our

results suggest that the valence signal was filtered out when it transfers from IC to dmPFC (MP-dmPFC) to M1. This description is given in Line 110-120. See also description Line 32-35.

8) L114: “Euclidian distance in all PC dimensions”: this should be identical to Euclidian distance w.o. PCA...And if this is the main conclusion, why is PCA even necessary?

Our response: We fully agree that PCA is not necessary in this case. We used PCA simply because the trajectory data is based on the first two dimensions of PCA. To match this data, we used PCA's dimensions. So, the PCA is not necessary, but it is a nice way to show the readers where the data derived from.

9) Fig3: Did you have control animals w.o. GtACR expression but with laser? How do you exclude the possibility that mice did not lick because they were startled by blue light? In addition, the power of blue light is not described in Methods.

Our response: We have considered this potential factor that affects mouse behavior. To minimize the effect from laser, we used sleeves (**Reviewer Figure 2**). Apparently, laser itself does not affect mouse behavior. Nevertheless, we still conducted another experiment that shines the laser in the middle phase during the persistent licking (**Fig. 3d, e**). No significant change was found in lick frequency, facial activity, and locomotor activity. Additionally, we measured the facial activity in the initial phase and did not find a significant increase of facial activity (**Supplementary Fig 9c**). Moreover, chemo-genetics results (**Supplementary Fig. 8b-d**) also suggest that the abnormal behavior is due to the silencing of MP neurons. Therefore, we excluded the possibility that mice did not lick because they were startled by the blue laser. The laser power we used is 5 mW for optogenetic silencing. It was added to the Methods. The previous work has demonstrated that even the laser power as high as 8 mW did not affect licking behavior¹⁶.

Reviewer Figure 2. Representative images showing mice were manipulated with (left) and without laser

10) Fig4: It is possible that IC/M1 activity changed because animals did not lick due to MP-dmPFC manipulation. Then, this could be an indirect effect. Can authors analyze trials with and without a lick separately (and also align activity to the first lick)? In addition, this requires control of mice without GtACR expression (see comment 9).

Our response: It is true that animals did not lick is due to the inactivation of MP-dmPFC. However, it should not be the animal's response to the laser itself (indirect effect) as we explained in comment 9.

It is not possible to align activity to the first lick in the trial without a lick. However, based on the coding properties of M1 and IC, it must be true that no increased activity in IC and M1 in the trials without lick. As we explained in the model (Figure 6), MP-dmPFC induced persistent activity in M1, while the persistent activity in IC is an indirect effect.

Since different mice perform differently, the experiment with control mice only gives rise to a large variation in the initiation bias which provides limited information. This corresponds to the reviewer's suggestion in the first time.

11) L186: Why does decreasing the number of neurons model the silencing of MP-dmPFC neurons? Isn't it more appropriate to inject a negative current into all neurons?

Our response: The principle of GtACRs is to temporally close the membrane channel. This causes GtACRs expressed cell temporally inactivated. It will further inhibit the communication between GtACRs expressed cell and other cells. Therefore, it is reasonable to assume that silencing of MP neuron is equivalent to deletion of MP neuron. In the experiment, even though we inject virus into motor cortex to label MP neurons, only a part of MP neurons can be labeled, because the area of motor cortex is large, and the virus cannot infect the whole area of motor cortex. Therefore, decreasing the number of neurons is adequate to simulate the effects of optogenetic silencing.

12) L193: Behavior in the persistent lick task is persistent in the sense that it lasts for many seconds, but the mechanism of persistence is the duration of sensory input, according to this sentence (i.e., nothing to do with mechanisms in the brain). Is this the take-home message? If not, the authors should rephrase it.

Our response: We have rephrased it as 'a successive sensory stimulus acts as an input signal for the dmPFC MP neurons'. Please see this correction in the abstract. In summary this brief successive input triggers dmPFC MP neurons, which further promote a persistent movement.

13) L199: What is the logic for excluding "motor command"?

Our response: If the MP-dmPFC represents the motor command, then the inactivation of MP-dmPFC should cause changes in the licking frequency in the middle phase, which is not the case (**Fig. 3h, i**). Therefore we concluded that MP-dmPFC were not involved in the direct control of tongue movement. So, we exclude the possibility that MP-dmPFC represent motor command.

Minor comment

14) Line 174: What is an "MP network"? Does it mean a network of MP-dmPFC neurons?

Our response: Yes, it includes MP-dmPFC excitatory neurons and some inhibitory neurons.

Reference

1. Churchland MM, *et al.* Neural population dynamics during reaching. *Nature* **487**, 51-56 (2012).
2. Linden H, Petersen PC, Vestergaard M, Berg RW. Movement is governed by rotational neural dynamics in spinal motor networks. *Nature* **610**, 526-531 (2022).

3. Inagaki HK, Fontolan L, Romani S, Svoboda K. Discrete attractor dynamics underlies persistent activity in the frontal cortex. *Nature* **566**, 212-217 (2019).
4. Yttri EA, Dudman JT. Opponent and bidirectional control of movement velocity in the basal ganglia. *Nature* **533**, 402-406 (2016).
5. Hughes JE, Amyx H, Howard JL, Nanry KP, Pollard GT. Health effects of water restriction to motivate lever-pressing in rats. *Lab Anim Sci* **44**, 135-140 (1994).
6. Bekkevold CM, Robertson KL, Reinhard MK, Battles AH, Rowland NE. Dehydration parameters and standards for laboratory mice. *J Am Assoc Lab Anim Sci* **52**, 233-239 (2013).
7. Rowland NE. Food or fluid restriction in common laboratory animals: balancing welfare considerations with scientific inquiry. *Comp Med* **57**, 149-160 (2007).
8. Guo ZV, *et al.* Procedures for behavioral experiments in head-fixed mice. *PLoS One* **9**, e88678 (2014).
9. Ivy CM, Scott GR. Evolved changes in breathing and CO₂ sensitivity in deer mice native to high altitudes. *Am J Physiol Regul Integr Comp Physiol* **315**, R1027-R1037 (2018).
10. Ivy CM, Scott GR. Control of breathing and the circulation in high-altitude mammals and birds. *Comp Biochem Physiol A Mol Integr Physiol* **186**, 66-74 (2015).
11. Hamad N, Travis SP. Weight loss at high altitude: pathophysiology and practical implications. *Eur J Gastroenterol Hepatol* **18**, 5-10 (2006).
12. Fusch C, Gfrorer W, Koch C, Thomas A, Grunert A, Moeller H. Water turnover and body composition during long-term exposure to high altitude (4,900-7,600 m). *J Appl Physiol (1985)* **80**, 1118-1125 (1996).
13. Xu D, *et al.* Cortical processing of flexible and context-dependent sensorimotor sequences. *Nature* **603**, 464-469 (2022).
14. Wang Y, Sun QQ. A long-range, recurrent neuronal network linking the emotion regions with the somatic motor cortex. *Cell Rep* **36**, 109733 (2021).
15. Wang L, *et al.* The coding of valence and identity in the mammalian taste system. *Nature* **558**, 127-131 (2018).

16. Otis JM, *et al.* Prefrontal cortex output circuits guide reward seeking through divergent cue encoding. *Nature* **543**, 103-107 (2017).

REVIEWERS' COMMENTS

Reviewer #4 (Remarks to the Author):

The authors addressed all of my concerns and significantly improved the manuscript.